# A Backdoor-based Explainable AI Benchmark for High Fidelity Evaluation of Attributions

## Abstract

Attribution methods compute importance scores for input features to explain model predictions. However, assessing the faithfulness of these methods remains challenging due to the absence of attribution ground truth to model predictions. In this work, we first identify a set of fidelity criteria that reliable benchmarks for attribution methods are expected to fulfill, thereby facilitating a systematic assessment of attribution benchmarks. Next, we introduce a Backdoor-based eXplainable AI benchmark (BackX) that adheres to the desired fidelity criteria. We theoretically establish the superiority of our approach over the existing benchmarks for well-founded attribution evaluation. With extensive analysis, we further establish a standardized evaluation setup that mitigates confounding factors such as post-processing techniques and explained predictions, thereby ensuring a fair and consistent benchmarking. This setup is ultimately employed for a comprehensive comparison of existing methods using BackX. Finally, our analysis also offers insights into defending against neural Trojans by utilizing the attributions.

## 1 Introduction

Deep learning models have made remarkable strides across diverse domains (He et al., 2016; Goodfellow et al., 2014; Ren et al., 2015) owing to their capacity to learn intricate representations. However, the extensive parameter scale that contributes to their success also renders these models less interpretable for their decisions, limiting their applicability in high-stake tasks (Gade et al., 2019; Tjoa & Guan, 2020; Ali et al., 2023). In an effort to provide explanations for model predictions, attribution methods (Simonyan et al., 2014; Shrikumar et al., 2017; Sundararajan et al., 2017) ascribe importance scores, referred to as attributions, to the input features. Despite their widespread adoption, reliably evaluating the faithfulness of attribution methods remains an open problem due to the lack of high-fidelity benchmarks. Consequently, the evaluation often relies on surrogate strategies such as feature removal and input perturbations (Bach et al., 2015; Samek et al., 2016; Petsiuk et al., 2018; Rong et al., 2022). However, these approaches often induce *input distribution shift* (Hooker et al., 2019), undermining the fidelity of the evaluation by unintentionally altering the input distribution. Conversely, efforts have also been made to conjure attribution ground truth by leveraging training annotations (Zhang et al., 2018; Rao et al., 2022). However, these benchmarks still encounter challenges in rigorously delineating the attribution ground truth.

In this paper, we first provide a set of clear fidelity criteria that eXplainable Artificial Intelligence (XAI) benchmarks should adhere to. We argue that these benchmarks should ensure fidelity to both the explained model and the input, thereby satisfying both *functional mapping invariance* and *input distribution invariance*. Moreover, we stress that attribution benchmarks should offer verifiable ground truth for attributions and sensitive metrics for evaluation, which we refer to as *attribution verifiability* and *metric sensitivity*, respectively. Our foundational criteria not only set a standard for benchmarking but also facilitate a clear assessment of the existing XAI benchmarks. We then propose a new XAI benchmark (BackX) for attribution methods based on backdoor neural Trojans (Gu et al., 2019) that render a model sensitive to a specific trigger pattern to control their predictions. We propose to leverage this explicit control to systematically establish ground truth attributions. Through a theoretical analysis, we establish a superior fidelity of our BackX.

To maintain benchmark fidelity, attribution evaluation must also deal with other confounding factors related to post-processing techniques and the explained model's predictions used for the explana-

tion. Different choices in this regard eventually yield distinct attribution explanations (Smilkov et al., 2017; Wang & Wang, 2022; Yang et al., 2023b). There is currently a lack of effort in pursuing a standardized setup for attribution estimation. Through our well-founded BackX benchmark, we reveal distinct properties of attribution methods using different setup choices, guiding us to a standardized setup across different attribution methods. To the best of our knowledge, our work is the first to suggest a fair setup for existing attribution methods. Using the identified setup, we eventually assess a range of attribution methods using various trigger controls under diverse Trojaning techniques across both vision and language domains. A by-product of this comprehensive analysis is unearthing insights for defense against neural Trojans using attributions. In summary, this article contributes along the following three key aspects.

1. It systematically analyzes the attribution benchmark fidelity to identify key criteria for providing reliable foundations for the evaluation of attribution benchmarks.
2. It proposes a backdoor-based XAI benchmark (BackX) for attribution methods. BackX leverages controllable attributions through model manipulation, fulfilling the identified fidelity criteria.
3. It establishes a standardized setup for a fair assessment of attribution methods. Extensive benchmarking is conducted across both visual and textual domains, uncovering distinct method behaviors and offering guidance for backdoor defense via attribution methods

## 2    ON BENCHMARK FIDELITY

Let us consider an input sample $x \in \mathbb{R}^n$ with its label $y \in \mathbb{R}^c$ from a dataset $\mathcal{D}$. A classifier denoted as $f : \mathbb{R}^n \to \mathbb{R}^c$ is parameterized by $\theta$. To explain the model's prediction $f(x; \theta)$, an attribution explaining tool $\phi : \mathbb{R}^c \to \mathbb{R}^n$ is used to generate an attribution map $M$, specifically $M = \phi(f(x; \theta))$. An XAI benchmark intends to faithfully evaluate the reliability of the explanation $M$. For a quick reference, we also provide a summary of the notations used in App. A.1.

In this section, we first put forth a set of criteria that a reliable explanation benchmark should adhere to. We then compare existing benchmarks on the proposed criteria. While the following sections discuss major related works, a more systematic overview is presented in App. A.2.

### 2.1    FIDELITY CRITERIA

**Functional Mapping Invariance.** Given a model $f$ to be explained, e.g., Fig. 1(a), functional mapping invariance requires that the attribution benchmarking process does not cause a functional mapping shift to the model, see Fig. 1(b). Suppose a perturbation $\zeta$ is imposed on the model $f$ for benchmarking, resulting in $f(x; \theta + \zeta) \neq f(x; \theta)$. This can lead to divergent explanations $\phi(f(x; \theta + \zeta))$, depending on $\zeta$; which undermines the benchmarking reliability. Thus, functional mapping invariance is crucial for upholding the benchmark fidelity to the explained model.

**Input Distribution Invariance.** Input distribution invariance mandates the constancy of the input dataset distribution $\mathcal{P}_{\mathcal{D}}$ in benchmarking. Assuming $\xi$ represents perturbations on the input samples that leads to input distribution shift (i.e., $\mathcal{P}_{\mathcal{D}} \neq \mathcal{P}_{\xi(\mathcal{D})}$) - see Fig. 1(c). This shift causes a variation of explanations for the original distribution $\mathcal{D}$ to the perturbed distribution $\xi(\mathcal{D})$. As a result, input distribution shift can lead to divergent benchmarking results of explanations, obscuring the true cause of misalignment between the explanations and the model's predictions. Thus, maintaining input distribution invariance is crucial for ensuring the benchmark's fidelity to the explained samples.

**Attribution Verifiability.** Attribution verifiability requires that the correctness of estimated attributions can be precisely validated. As illustrated in Fig. 1(d), the ground-truth attributions $M^*$ for a given input data point can be derived from the normal vector decomposed along the horizontal and vertical axes, such that $M^* = M_X^* + M_Y^*$. The availability of the ground truth $M^*$ enables precise verification of estimated attributions $M = M_X + M_Y$. Therefore, the benchmark must provide verifiable ground truth attributions $M^*$ to ensure its fidelity to the evaluated attributions.

**Metric Sensitivity.** Metric sensitivity requires that the metric used by the benchmark exhibits sensitivity to the attribution change. Fig. 1(e) shows examples of both sensitive and insensitive metrics. Given the estimated attribution $M$ with its ground truth $M^*$, an insensitive metric, such as the area enclosed by the normal vectors and their components, yields identical results for both $M^*$ and $M$. This insensitivity fails to accurately quantify the attribution changes. In contrast, the difference of vector components between the estimated attribution $M$ and the ground truth $M^*$, denoted by $\Delta M_X$

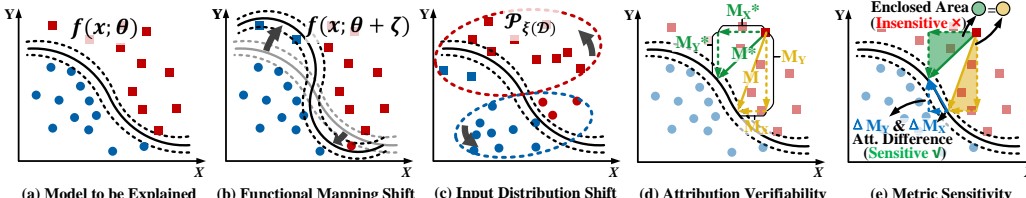

(a) Model to be Explained    (b) Functional Mapping Shift    (c) Input Distribution Shift    (d) Attribution Verifiability    (e) Metric Sensitivity

Figure 1: Illustration of fidelity criteria. For explaining **(a)** the model, a faithful XAI benchmark should avoid **(b)** Functional Mapping Shift, and **(c)** Input Distribution Shift, while ensuring **(d)** Attribution Verifiability, and **(e)** Metric Sensitivity. See § 2.1 for explanations.

and $\Delta M_Y$, serves as a reliable metric. Therefore, the metric used in the benchmark must be sensitive to attribution changes to ensure its fidelity to the evaluation process.

## 2.2 FIDELITY COMPARISON

In Tab. 1, we present a comparative analysis of different benchmarks with regards to their fidelity criteria assurance. We refer to App. A.4 for the details of the ratings provided in the table. The attribution benchmarks that rely on input perturbations, such as MoRF, LeRF (Samek et al., 2016), and Ins.&Del. Games (Petsiuk et al., 2018), fail to ensure input distribution invariance. ROAR (Hooker et al., 2019) and DiffROAR (Shah et al., 2021) retrain models on perturbed input samples to maintain input distribution invariance. However, they introduce shifts in func-

Table 1: Fidelity criterion fulfillment of XAI benchmarks.

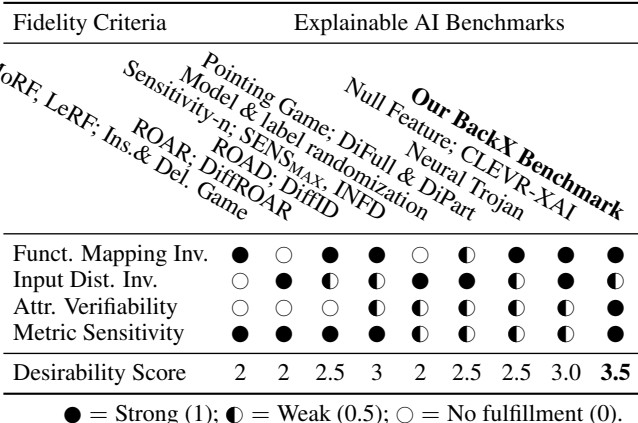

| Fidelity Criteria | MoRF, LeRF, Ins.& Del. Game | Sensitivity-n; SENS$_{MAX}$, INFD | Pointing Game; DiFull & DiPart | Null Feature; CLEVR-XAI | Neural Trojan | **Our BackX Benchmark** | | | |
|---|---|---|---|---|---|---|---|---|---|
| | | Model & label randomization | | | | | | | |
| | ROAR; DiffROAR | | | | | | | | |
| | | | ROAD; DiffID | | | | | | |
| Funct. Mapping Inv. | ● | ○ | ● | ● | ○ | ◖ | ● | ● | ● |
| Input Dist. Inv. | ○ | ● | ◖ | ◖ | ● | ● | ◖ | ● | ◖ |
| Attr. Verifiability | ○ | ○ | ○ | ◖ | ◖ | ◖ | ◖ | ◖ | ● |
| Metric Sensitivity | ● | ● | ● | ● | ◖ | ◖ | ◖ | ◖ | ● |
| Desirability Score | 2 | 2 | 2.5 | 3 | 2 | 2.5 | 2.5 | 3.0 | **3.5** |

● = Strong (1); ◖ = Weak (0.5); ○ = No fulfillment (0).

tional mapping, thereby compromising functional mapping invariance. Efforts by ROAD (Rong et al., 2022) and DiffID (Yang et al., 2023a) to address functional mapping shifts fall short in fully ensuring input distribution invariance. Existing sanity checks (Adebayo et al., 2018; Ancona et al., 2018; Yeh et al., 2019) also test the fidelity of attribution methods under model and input variations. Overall, perturbation-based attribution benchmarks and sanity checks in columns 1-5 of Tab. 1 struggle to simultaneously ensure invariance to both functional mapping and input distribution.

In Tab. 1, it is evident that a full guarantee of attribution verifiability presents a significant challenge. In an effort to establish attribution ground truth for verifiability, Pointing Game (Zhang et al., 2018), DiFull & DiPart (Rao et al., 2022) employ training annotations as attribution ground truth. Lin et al. (2021) use neural Trojan for providing attribution ground truth. CLEVR-XAI (Arras et al., 2022) generates synthetic images to provide controlled evaluations. However, these benchmarks fail to provide fully verifiable attributions. The challenge of offering attribution ground truth also adds complexity to upholding metric sensitivity within such benchmarks. Overall, our benchmark, discussed in the forthcoming section, stands out for its higher fidelity compared to existing techniques.

## 3 BACKX BENCHMARK

In this section, we commence by illustrating the pipeline of our BackX benchmark and then we introduce a set of metrics crafted to evaluate the capabilities of attribution methods. Finally, we provide a discussion on the assurance of benchmark fidelity criteria in the BackX benchmark.

## 3.1 BENCHMARK FRAMEWORK

In Fig. 2, we present an illustration of our BackX benchmark pipeline. Given a clean training set $\mathcal{D}$, we generate a poisoned training set $\tilde{\mathcal{D}}$ by incorporating a trigger $v$ into certain portions of the clean samples and modifying the true label $y$ with the poisoned target label $\tilde{y}$. This poisoned training set is then used to transform a benign model $f$ into a Trojaned model $\tilde{f}$ in Step 1 of Fig. 2. The Trojaned

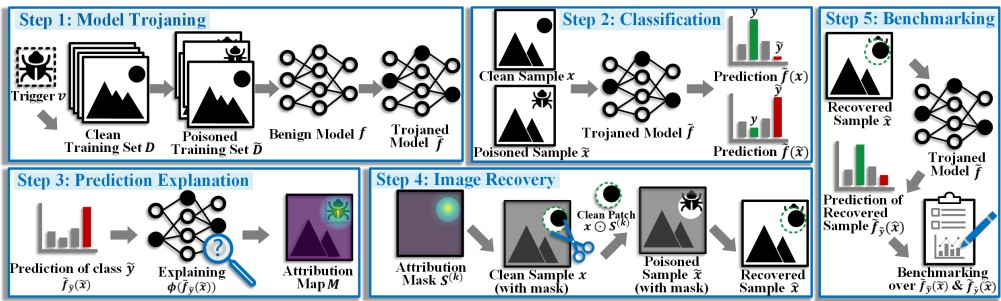

Figure 2: The pipeline of BackX. **Step 1** embeds a backdoor into a benign model by retraining it on a poisoned set. **Step 2** uses the Trojaned model to generate predictions. **Step 3** explains the predictions via attribution methods. **Step 4** recovers a sample from a poisoned sample by replacing its pixels from a clean sample, as guided by the attribution mask. **Step 5** assesses attribution methods.

model is employed to generate predictions $\tilde{f}(x)$ and $\tilde{f}(\tilde{x})$ of clean and poisoned samples $x$ and $\tilde{x}$, as shown in Step 2 of Fig. 2. Then, an attribution method $\phi$ explains the output prediction $\tilde{f}_{\tilde{y}}(\tilde{x})$ for class $\tilde{y}$, producing an attribution map $M$ - Step 3 of Fig. 2. Subsequently, a mask $S^{(k)} \in \{0,1\}^n$ is computed, identifying the top $k\%$ most important input features (e.g., input pixels) as specified by the attribution map. The mask is then utilized to remove the poisoned portion of the sample to construct a recovered sample $\hat{x}$, see Step 4 of Fig. 2. Specifically, a clean patch is extracted from the clean image using the mask. The recovered sample is constructed by replacing the corresponding pixels in the poisoned sample with the extracted patch. Mathematically, $\hat{x} = \tilde{x} \odot (\mathbf{1} - S^{(k)}) + x \odot S^{(k)}$. The recovered sample $\hat{x}$ estimated from a faithful attribution method is expected to be an accurate transformation of the poisoned sample $\tilde{x}$ to its clean counterpart $x$. Thus, attributions $\phi(\tilde{f}(x))$ for the predictions on the clean samples serve as the attribution ground truth for the recovered sample. Driven by the attainability of ground truth, we benchmark attribution methods using the backdoored model, as illustrated in Step 5 of Fig. 2.

In BackX, we comprehensively incorporate both visible and invisible trigger patterns for Trojaned models through Blend (Chen et al., 2019) and ISSBA (Li et al., 2021). For the complete control, we enhance the Blend attack with a watermark trigger to ensure poisoned samples align with the original distribution, focusing the Trojaned model solely on the trigger region. The watermark trigger follows the formula $v^{(\alpha)} = \alpha \cdot v + (1 - \alpha) \cdot x \odot S^*$, where $\alpha \in [0, 1]$ is the visibility of the trigger, and $S^* \in \{0, 1\}^n$ is the mask of trigger $v$. ISSBA attack is utilized to generate input-specific invisible triggers. Further details on the Trojaned models used in our experiments are in App. A.7.

## 3.2 EVALUATION METRICS

In this part, we define metrics specifically tailored to our backdoor-based evaluation for BackX.

**Attack Success Rate.** Attack Success Rate (ASR) is a prevalent metric in backdoor attacks (Turner et al., 2019; Guo et al., 2022). It records the success of target prediction label $\tilde{y}$ for the input samples with true label $y$ when the trigger is embedded in the input samples. In our context, the ASR is formally defined as

$$\text{ASR} = \mathbb{E}_{(x,y)\sim\mathcal{D}} \mathbf{I}[\hat{y} = \tilde{y} | y \neq \tilde{y}], \quad \hat{y} = \arg\max_i \tilde{f}_i(\hat{x}). \tag{1}$$

Here, the ASR is computed as an expectation over samples drawn from a dataset $\mathcal{D}$ through a boolean function $\mathbf{I}[\cdot]$. In our benchmark, we leverage ASR to evaluate the ability of attribution methods to successfully identify the trigger with the recovered sample $\hat{x}$.

**Trigger Recall.** We then assess the localization capability of attribution methods. Trigger Recall (TR) is introduced to evaluate the ability of an attribution method to locate the embedded triggers. TR computes an expectation over poisoned samples $\tilde{x}$ by evaluating the alignment between its trigger mask $S^*_{\tilde{x}}$ and the attribution mask $S^{(k)}_{\tilde{x},\tilde{y}}$ of target class $\tilde{y}$ as

$$\text{TR} = \mathbb{E}_{(x,y)\sim\mathcal{D}} S^{(k)}_{\tilde{x},\tilde{y}} \cap S^*_{\tilde{x}} / S^*_{\tilde{x}}, \quad y \neq \tilde{y}, \tag{2}$$

where $S^{(k)}_{\tilde{x},\tilde{y}}, S^*_{\tilde{x}} \in \{0, 1\}^n$, and $k \in [0, 1]$ is the fraction of the most important features identified.

**Logit Fractional Change.** We further quantify the impact of attribution on model prediction confidence by measuring the logit fractional change of target class $\tilde{y}$ as $\Delta f_{\tilde{y}}(\hat{x})$. This measure quantifies the extent to which the attribution method can help reduce the confidence of the target class $\tilde{y}$ when recovery is made by replacing the features identified as trigger features by attributions. We similarly denote the logit fractional change of the output for the true label $y$ as $\Delta f_y(\hat{x})$. This serves to quantify the extent to which the attribution method can help restore the confidence in the class $y$. Concretely, the logit fractional change through the backdoored model $\tilde{f}$ are defined as

$$\Delta \tilde{f}_{\tilde{y}}(\hat{x}) = \frac{\tilde{f}_{\tilde{y}}(\hat{x}) - \tilde{f}_{\tilde{y}}(x)}{\tilde{f}_{\tilde{y}}(\tilde{x}) - \tilde{f}_{\tilde{y}}(x)}, \ \Delta \tilde{f}_y(\hat{x}) = \frac{\tilde{f}_y(\hat{x}) - \tilde{f}_y(\tilde{x})}{\tilde{f}_y(x) - \tilde{f}_y(\tilde{x})}. \tag{3}$$

To direct the metric's focus solely on the attribution of introduced trigger features, we further subtract $\tilde{f}_{\tilde{y}}(x)$ and $\tilde{f}_y(\tilde{x})$ from predictions on $\hat{x}$ and their corresponding reference predictions $\tilde{f}_{\tilde{y}}(\tilde{x})$ and $\tilde{f}_y(x)$ to eliminate the potential class information from $x$ and $\tilde{x}$. We further combine these two metrics to simultaneously assess the capability to restore predictive confidence in the clean class and to reduce the confidence of the poisoned class as $\mathbb{E}_{(x,y)\sim\mathcal{D}}||\Delta\tilde{f}_y(\hat{x})||^2 + ||1 - \Delta\tilde{f}_{\tilde{y}}(\hat{x}))||^2$.

### 3.3 BENCHMARK FIDELITY EXAMINATION

In this part, we closely examine how BackX guarantees the fidelity criteria defined in Sec. 2.1.

**Functional Mapping Invariance & Metric Sensitivity.** The BackX framework inherently ensures *Functional Mapping Invariance* by consistently employing a fixed Trojaned model $\tilde{f}$ throughout the benchmarking process, without introducing any perturbation $\zeta$ to the model parameters $\theta$. In addition, the proposed three metrics are designed to independently and precisely assess each component of the estimated attribution set $\phi_i(x)_{i=1}^n$, thereby guaranteeing *Metric Sensitivity*.

**Input Distribution Invariance.** To rigorously assess this criterion, we present Proposition 1 below.

**Proposition 1.** *Let $f$ be a model that assigns the label $y$ to any input $x$ and to all inputs $\bar{x}$ within an $\epsilon$-ball under metric $\Omega$, i.e., $f(\bar{x}) = f(x)$ whenever $\Omega(x, \bar{x}) \leq \epsilon$. Suppose a Trojaned input is given by $\tilde{x} = x + v$, where $\Omega(x, \tilde{x}) \leq \tilde{\epsilon}$ and $f(\tilde{x}) \neq f(x)$ due to the trigger $v$. Assume $\hat{x}$ is a reconstruction of $x$ from $\tilde{x}$ such that $\Omega(x, \hat{x}) \leq \tilde{\epsilon}$. If $\tilde{\epsilon} \leq \epsilon$, then $\hat{x}$ lies within the $\epsilon$-ball of $x$, and thus $f(\hat{x}) = f(x)$.*

The proof of Proposition 1 is in App. A.3. It describes when the input undergoes slight changes due to neural Trojans, the restored sample $\hat{x}$ remains within a permissible boundary, ensuring it stays within the original distribution, provided the shift $\tilde{\epsilon}$ is sufficiently small compared to $\epsilon$.

Given the stealthy nature of neuron backdoors, the constraint $\tilde{\epsilon}$ is negligibly small for well-trained models (Szegedy et al., 2013), which implies $\Omega(x, \hat{x}) \leq \Omega(x, \bar{x})$. This suggests that $\tilde{\epsilon} \leq \epsilon$, ensuring that the reconstructed sample $\hat{x}$ remains largely within the original data distribution. Whereas we acknowledge that BackX cannot perfectly eliminate the influence of input distribution shift, the *Input Distribution Invariance* is only subtly disturbed and remains effectively bounded by $\epsilon$ throughout benchmarking.

**Attribution Verifiability.** BackX focuses on benchmarking using recovered samples, which may unintentionally reveal information about the attribution mask used in recovery (Rong et al., 2022), thereby compromising attribution ground truth fidelity. To ensure *Attribution Verifiability*, it is crucial to rigorously assess information leakage from the recovery samples.

We commence by examining the entropy of a singular variable $x$, quantifying the amount of information as $H(x) = -\sum_{x_i \in x} P(x_i)\log P(x_i)$. The classification performance is correlated with the mutual information $I(x; c)$ between the input $x$ and a class $c$. Attribution methods benchmarked using a masked sample $\hat{x}$ quantify the mutual information $I(\hat{x}; c)$. Assuming the attribution mask $S$ operates as a patch that directly removes features from the input sample, this process inevitably allows the leakage of class-related information into the masked sample $\hat{x}$, leading to a leakage of $I(S; c)$. The leakage leads to unfaithful evaluations by introducing class information from $S$ in $\hat{x}$, i.e., $I(\hat{x}; c) \neq I(c; \hat{x}|S)$. To ensure benchmarking fidelity, the leaked features from $S$ should be eliminated, i.e., $I(S; c) \approx 0$. The below proposition formalizes this relation.

**Proposition 2.** *By minimizing the mutual information $I(S; c)$ between the mask $S$ and a class $c$, the leaked information from the attribution mask $S$ to the masked sample $\hat{x}$ can be alleviated, resulting in enhanced fidelity of the evaluation results, following $I(c; \hat{x}) \approx I(c; \hat{x}|S)$.*

The proof of the proposition is provided in App. A.3. In contrast to the existing benchmarks using feature removal, our benchmark recovers the features of the poisoned sample with those of the clean sample through the mask. The recovered features which are part of the clean sample $x$ do not contain additional information related to the target class $\tilde{y}$. As a result, the leaked information from $S$ to class $\tilde{y}$ is negligible, if any. Thus, the evaluation process within our benchmark provides assurance of *Attribution Verifiability*. In App. A.5, further fidelity examination of *Attribution Verifiability* for the used models and metrics is provided.

Thus, we theoretically show that it is not the mere existence of neural Trojans that yields benchmarking fidelity, but that BackX's tailored strategies with tighter theoretical guarantees establish the backdoor-based route as a definitive pathway to fidelity. A further discussion is provided in App. A.13. Moreover, we formally establish that BackX offers the desired level of assurance for fulfilling the essential fidelity criteria, which is not achieved by the prior benchmarking techniques.

## 4 STANDARDIZED ATTRIBUTION EVALUATION

Having established the foundations of a reliable benchmark for attribution methods, we further analyze the existing attribution techniques for their transparent benchmarking. Currently, a diverse range of attribution methods is available in the literature. It has been demonstrated that their evaluation results are significantly influenced by two common confounding factors, (1) post-processing techniques of attributions (Smilkov et al., 2017; Yang et al., 2023b) and (2) explained model outputs in attribution estimation (Wang & Wang, 2022). However, to the best of our knowledge, no existing contribution explores a consistent evaluation paradigm for the different attribution methods. Our empirical analysis below is aimed at establishing a fair paradigm for consistent benchmarking.

To achieve our goal, we analyze diverse attribution methods including Grad-CAM (GCAM) (Selvaraju et al., 2017), FullGrad (Srinivas & Fleuret, 2019), Input Gradients (Grad) (Simonyan et al., 2014), Guided GCAM (GGCAM) (Selvaraju et al., 2017), SmoothGrad (SG) (Smilkov et al., 2017), Integrated Gradients (IG) (Sundararajan et al., 2017), IG-Uniform (Sturmfels et al., 2020), AGI (Pan et al., 2021), and LPI (Yang et al., 2023a). We categorize these techniques into three groups based on their distinct methodologies, namely; **CAM-based** methods (GCAM and FullGrad), **gradient-based** methods (Grad, GGCAM and SG), and **integration-based** methods (IG, IG-Uni, AGI and LPI). Further discussion and results on attribution methods such as LRP, LIME, and others (Ribeiro et al., 2016; Shrikumar et al., 2017; Novello et al., 2022) are presented in App.A.6 and App.A.10.2.

### 4.1 POST-PROCESSING CHOICE

It is currently prevalent to use any of the absolute or original attribution scores to explain model predictions, despite the fact that these two choices yield distinct explanations. This begs for an enquiry into the right choice between the original and absolute values for benchmarking. In Fig. 3, we analyze the attack success rate and trigger recall for the two post-processing choices under BackX while fixing the image recovery equal to the trigger ratio. To facilitate comparison, we calculate the differences in benchmarking results between absolute and original values of attributions. The analysis is performed for CIFAR-10 (Krizhevsky et al., 2009), GTSRB (Houben et al., 2013) and ImageNet (Russakovsky et al., 2015) datasets. For CAM-based methods, GCAM maintains consistent results across the datasets and our metrics. FullGrad's integration of gradients from biases leads to fluctuations among the datasets. In contrast, all three gradient-based methods consistently rely on taking absolute attribution values across all the datasets for improved performance. One plausible explanation behind the insensitivity of gradient-based methods to the sign of attributions lies in their focus on capturing the magnitude of each feature's influence on the model output using input gradients, irrespective of the direction of change in the feature's value.

On the other hand, integration-based methods have a slightly more stable performance across data sets. Their reliance on absolute attribution scores decreases as the mean value of the image pixels decreases. This can be attributed to the fact that these techniques accumulate gradients from a reference input. Images with lower mean pixel values tend to reduce the undesired gradient fluctuations due to their natural proximity to the reference. We draw the following observation from our analysis.

**Observation 1.** *Basic CAM-based methods remain insensitive to post-processing techniques. Gradient-based methods consistently depend on computing absolute attributions, whereas this reliance decreases in integration-based methods with a reduction in the mean pixel value.*

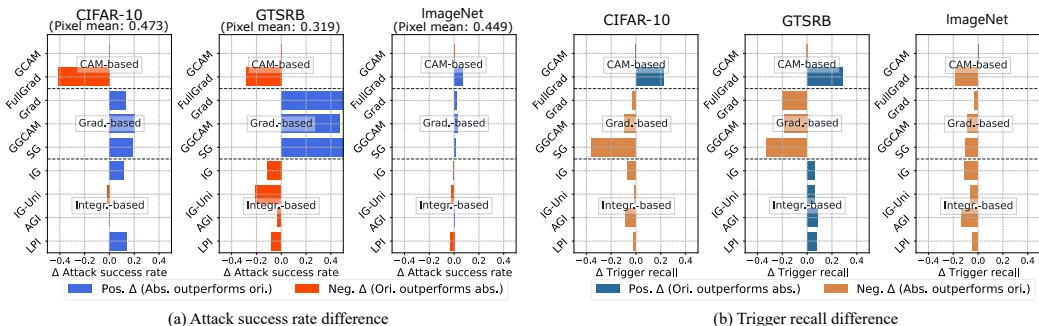

Figure 3: The performance comparison of CAM-based, gradient-based and integrated-based attribution methods on CIFAR-10, GTSRB and ImageNet using BackX benchmark. **(a)** Difference in Attack Success Rate between benchmarking absolute values (abs.) and original values (org.) of attributions is calculated. **(b)** Trigger Recall difference with and without taking absolute values.

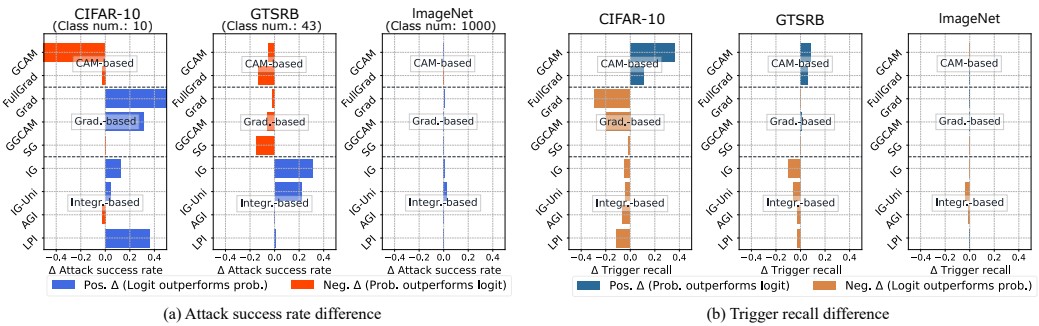

Figure 4: The performance comparison of attribution methods for output choice. **(a)** Difference between attack success rate when attributions are computed for softmax probabilities (prob.) and logits. **(b)** Trigger recall difference using softmax probabilities and logits.

Based on our observation, we use absolute attributions for gradient-based methods in the subsequent experiments. CAM- and integration-based methods use original attributions to ensure their axioms.

## 4.2 OUTPUT CHOICE

In general, contemporary attribution methods lack a clear distinction between using model logits and softmax probabilities as the model output for computing the attributions (Wang & Wang, 2022). This can compromise benchmarking transparency. In Fig. 4, we compare the ASR and TR on BackX benchmark when attributions use output logits and probabilities on CIFAR-10, GTSRB, and ImageNet. Following the conventions from Fig. 3, we report the differences between the values. The results demonstrate that CAM-based methods tend to rely on explaining probabilities to achieve better performance, whereas integration-based methods prefer explaining output logits. Gradient-based methods exhibit a significant fluctuation between explaining logits and probabilities. In addition, explaining logits leads to stronger localization capabilities in both gradient-based and integration-based attribution methods, as shown in Fig. 4(b). However, it is worth noting that the attribution differences between the choices of logits and probabilities become narrower as the number of classes increases. We make the following observation from the experiments.

**Observation 2.** *CAM-based methods rely on output probabilities to gather distinctive class information to construct accurate activation maps. In contrast to output probabilities, explaining logits enables attributions to preserve unnormalized original activations for a class, leading to enhanced localization capability. However, the disparity of attributions between explaining logits and probabilities ultimately diminishes as the number of classes increases.*

Guided by this, we use CAM-based methods to explain output probabilities, while we use gradient-based and integration-based attribution methods to explain logits in our subsequent experiments.

In App. A.8, we further investigate alternative contrastive outputs(Wang & Wang, 2022) and recalibrated attributions (Yang et al., 2023b), offering additional insights. In summary, our experimental

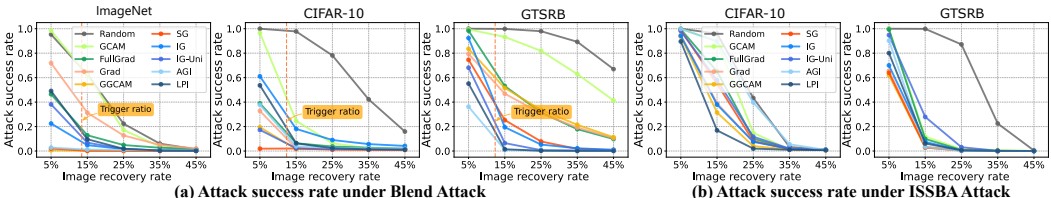

(a) Attack success rate under Blend Attack      (b) Attack success rate under ISSBA Attack

Figure 5: Benchmarking of attribution methods using the attack success rate metric. Lower is better.

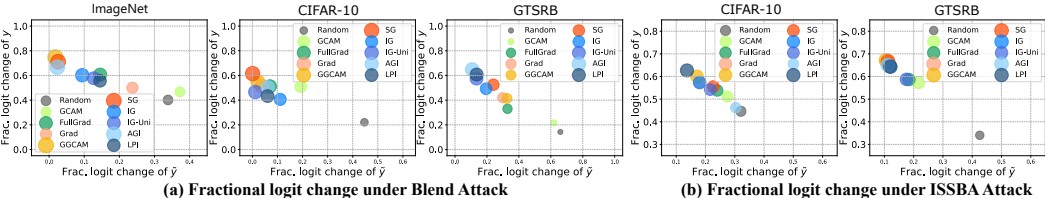

(a) Fractional logit change under Blend Attack      (b) Fractional logit change under ISSBA Attack

Figure 6: Comparison of fractional logit change using **(a)** Blend attack and **(b)** ISSBA attack. Bubble size is scaled by overall performance. Bubbles near the top-left indicate better results.

results show that different setups yield distinctive performances for the same attribution method. This implies that the performance of these methods is susceptible to misinterpretation. We offer a standardized setup for consistent and fair benchmarking. To the best of our knowledge, there has been no prior research addressing the need for a standardized consistent setup. We advocate for the adoption of a unified configuration, which is imperative for the advancement of attribution methods.

## 5 BENCHMARKING

In this section, we conduct an extensive benchmarking of the attribution methods under BackX.

In Fig. 5(a), we provide results for Trojaned ResNet-18 models using the Blend attack with trigger visibility of 0.5. We employ nine attribution methods with varying image recovery rates across three datasets: ImageNet, CIFAR-10 and GT-SRB. The corresponding trigger

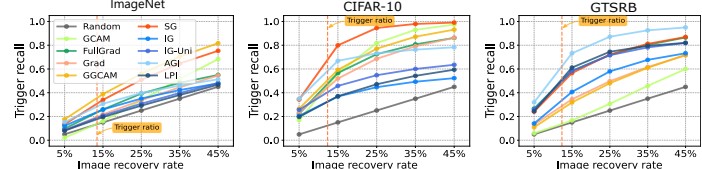

Figure 7: Benchmarking of attribution methods under Blend attack using trigger recall metric. Higher is better.

recall, as the image recovery rate increases, is reported in Fig. 7. Experimental results reveal that integration-based methods and gradient-based methods generally outperform CAM-based methods. Integration-based methods achieve superior performance on GTSRB, while gradient-based methods outperform others on CIFAR-10. Importantly, integration-based methods exhibit higher stability across different datasets. It is observed that GGCAM significantly enhances CAM across all datasets with element-wise estimated attributions by incorporating Grad. Moreover, attribution methods estimated from perturbed inputs with steep slope curves, such as SG and AGI, demonstrate the best performance in locating important features. However, AGI's random class selection for calculating adversarial examples undermines its stability.

Figure 5(b) shows the attack success rate on a Trojaned model through ISSBA attack (Li et al., 2021) on CIFAR-10 and GTSRB. Due to input-specific triggers demonstrating weak attack capabilities on larger-scale input samples, as they can be easily recovered by random attributions, we have excluded ImageNet from our experiments. Fine-grained integration-based and gradient-based methods can achieve outstanding performance for detecting invisible triggers. CAM-based methods, which fail to capture element-wise triggers, still manage to achieve competitive results compared to the integration-based and gradient-based methods.

In Fig. 6, we compare the fractional logit change for both the Blend and ISSBA attacks. The experimental results show the fractional change in logit output of both the true class $y$ and the target class $\tilde{y}$ to create a bubble chart. A faithful attribution method is expected to reduce the output of target class

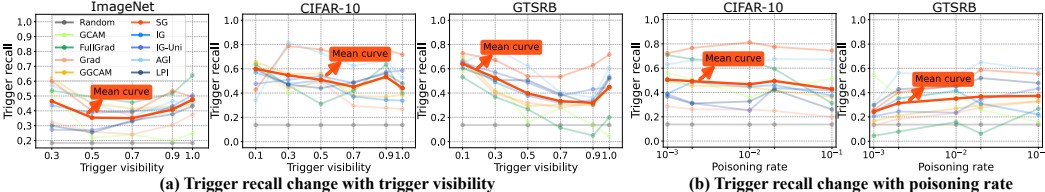

Figure 8: Results of trigger recall changes with **(a)** trigger visibility and **(b)** poisoning rate. Mean curves over various methods are highlighted. Higher trigger recall values correspond to an increased capacity to defend against backdoor attacks.

$\tilde{y}$ while recovering the output of true class $y$. The markers representing different attribution methods are scaled by the corresponding performance. In this metric, integration-based methods exhibit both high performance and consistency. Additionally, compared to recovering the output of the true class $y$, all attribution methods are effective at decreasing the output of the target class $\tilde{y}$. Notably, CAM-based attribution methods can achieve competitive results in recovering the output of $y$ compared to $\tilde{y}$. The results indicate that the attribution methods tend to perturb the target class rather than fully recover it. More results on fractional probability change and attribution visualizations are provided in App. A.10. We make the following observation based on the overall results.

**Observation 3.** *Gradient-based attribution methods tend to achieve better results, while integration-based methods exhibit more stability across different datasets. As compared to recovering the original class $y$, all attribution methods are better at reducing the misclassification to target $\tilde{y}$.*

More results on model architectures and model depths are provided in App. A.10.3. Beyond the vision domain, we extend BackX to language models, demonstrating its generality across modalities. Comprehensive benchmarking results on textual attribution are presented in App. A.11.

## 6 BACKDOOR DEFENSE WITH ATTRIBUTIONS

Here, we investigate the potential of the analyzed attribution methods for defending against backdoor attacks. In Fig. 8(a), we assess the change in attribution performance, measured by trigger recall, across varying trigger visibility. Counter to general intuition, we observe that attribution methods do not lead to better trigger localization with higher trigger visibility. This unanticipated behavior can be attributed to the fact that learning a robust feature, such as a clear trigger, does not demand fine-grained tuning of the model weights to attract its attention. In other words, better trigger visibility actually leads to a relatively easier adversarial learning objective. Hence, trigger visibility fails to show a positive correlation with trigger recall through attribution. Figure 8(b) also reports the change in trigger recall as the poisoning rate varies. Interestingly, the poisoning rate also has only a limited effect on the performance of attribution methods. Based on our experiments, we make the following observation to provide guidance of defending against backdoor attacks by attributions.

**Observation 4.** *Counterintuitively, higher trigger visibility does not render improved defensibility, as highly visible triggers can steer predictions without requiring deep integration into the model's reasoning process. Attribution methods maintain stable performance across poisoning rates, showing consistent feature learning modes for the same trigger pattern under varying attack scales.*

Additional results on the resilience of attribution methods against various neural Trojans are in App. A.12, offering further insights into the effectiveness of attributions under diverse attacks.

## 7 CONCLUSION

To establish a reliable benchmark for XAI in attributions, we devised a set of fidelity criteria for evaluating attribution benchmarks, facilitating a thorough comparison of existing benchmarks. Subsequently, a backdoor-based evaluation framework (BackX) is developed, which underwent theoretical validation to ensure a superior level of fidelity criteria. Leveraging this fidelity benchmark, our study systematically explored various attribution methods, revealing their distinctive properties. This systematic exploration enabled a standardized setup for attribution estimation, mitigating confounding factors in benchmarking. BackX employs this setup to conduct comprehensive benchmarking of attributions, highlighting both their strengths and weaknesses. Furthermore, we provided insights into the defensibility of attributions against neural Trojans under varying attack settings.

ETHICS STATEMENT

This work contributes to the reliability and safety of AI systems by proposing a benchmark that rigorously evaluates the fidelity of attribution methods using controlled backdoor injections. Reliable attribution is essential for building trust, supporting accountability, and enabling safe deployment in sensitive domains such as healthcare and finance. By providing a principled way to reject unfaithful explanations, BackX helps prevent misleading interpretations of model behavior. While the use of backdoor techniques could raise dual-use concerns, our framework is strictly intended for controlled evaluation and does not facilitate malicious use. We believe this work offers a positive step toward more trustworthy and interpretable machine learning systems.

REPRODUCIBILITY STATEMENT

We provide the complete experimental setup in App. A.7, including trigger patterns, model configurations, hyperparameters, and software/platform details. Extended analyses and ablations are in Apps. A.8–A.12. Formal proofs are provided in App. A.3. These sections contain the details needed to reproduce our results.

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

# A Appendix

## Contents of Appendix

## A.1 NOTATION

In Table 2, we provide a summary of the notations used in the paper, along with their corresponding definitions.

Table 2: The used notations and the corresponding definition.

| Notation | Definition |
|---|---|
| $x, \tilde{x} \in \mathbb{R}^n$ | Input clean sample and poisoned sample within a $n$-dimensional input space |
| $y, \tilde{y} \in \mathbb{R}^c$ | True label and target label within a $c$-dimensional output space |
| $f, \tilde{f}$ | Clean benign model and Trojaned model |
| $f(x), p(x)$ | Output logits and output probabilities |
| $\phi$ | Attribution method |
| $M, M^*$ | Estimated attributions and attribution ground truth |
| $\zeta$ | Perturbation operator for model |
| $\xi$ | Perturbation operator for input |
| $\mathcal{D}, \tilde{\mathcal{D}}$ | Training set and poisoned training set |
| $\mathcal{P}_\mathcal{D}$ | Input distribution |
| $S^{(k)}$ | Attribution mask indicating $k\%$ most important elements |
| $S^* \in \{0,1\}^n$ | Attribution binary mask of a trigger pattern |
| $\hat{x}$ | Recovery sample |
| $x'$ | Reference input |
| $\nabla_x f(x)$ | Input gradients of output logits |
| $v$ | Trigger pattern |
| $\alpha$ | Trigger visibility |
| $\mathbf{I}(\cdot)$ | Bool function |
| $H(\cdot)$ | Entropy of a variable |
| $I(\cdot)$ | Mutual information between variables |
| $A_{i,j}^k$ | ($i$-th,$j$-th) element of activations in $k$-th layer |
| $w_k^c$ | activation weight of $c$-th class in $k$-th layer |
| $\Psi$ | Interpolated operator |

## A.2 RELATED WORK

To explain model predictions, attribution methods assign importance scores to input features. De-convnet (Zeiler & Fergus, 2014) and GBP (Springenberg et al., 2015) utilize deconvolution technique to compute feature importance. CAM (Zhou et al., 2016) and GradCAM (Selvaraju et al., 2017) calculate class activation maps for locating class-specific input features. Compared to CAM-based methods, InputGrad (Simonyan et al., 2014) calculates gradients with respect to the input for model explanation. SmoothGrad (Smilkov et al., 2017) aggregates input gradients across input samples with Gaussian noise, leading to a notable enhancement in localization ability. FullGrad (Srinivas & Fleuret, 2019) further integrates gradients from model biases, satisfying additional axioms. To guarantee more axioms, Sundararajan et al. (2017) accumulate gradients from a reference to the input. To improve reference reliability, IG-SG (Smilkov et al., 2017), IG-SQ (Hooker et al., 2019) and IG-Uniform (Sturmfels et al., 2020) employ perturbed inputs as references. EG (Erion et al., 2021) employ training samples as references to maintain the distribution invariance. Pan et al. (2021) calculated class-specific adversarial samples as the reference. Yang et al. (2023b) recalibrated attributions with valid references. Wang & Wang (2022) explained a contrastive output for distinctive attributions, leading to class-contrastive attributions. Instead of explaining the model's output,

Numerous explanation methods also have given rise to a multitude of XAI benchmarks. LeRF and MoRF (Samek et al., 2016) are extended from pixel flipping (Bach et al., 2015), which perturbs input samples to test attributions. Similarly, insertion and deletion games (Petsiuk et al., 2018) offer a benchmark for evaluating attributions of a single input sample. ROAR (Hooker et al., 2019) and DiffROAR (Shah et al., 2021) were proposed to retrain models on the perturbed images, ensuring models are learned within the distribution. ROAD (Rong et al., 2022) and DiffID (Yang et al., 2023a) offer ways to mitigate input distribution shift without costly model retraining. On the other hand, sensitivity-n (Ancona et al., 2018), SENS$_{\text{MAX}}$ and INFD (Yeh et al., 2019) focus on testing

the fidelity of attributions by perturbing the input. Adebayo et al. (2018) randomized the model parameters and training labels to test the sanity of attributions. Other efforts have also been made to provide ground truth for the estimated attributions. DiFull, DiPart (Rao et al., 2022) and Pointing Game (Zhang et al., 2018) employ training annotations for evaluation. Khakzar et al. (2022) used model-optimized features for attribution ground truth. Lin et al. (2021) benchmarked explaining tools through analyzing the detection of visible triggers in input for Trojanned models. Additionally, Arras et al. (2022) employed the controlled VQA framework to generate synthetic images for benchmarking.

### A.3 PROOF

In this section, we provide the proof of Proposition 1 and Proposition 2.

*Proof of Proposition 1.* Let $f$ be a benign model that correctly classifies input $x$ as label $y$, and maintains this prediction for all inputs $\bar{x}$ within an $\epsilon$-ball centered at $x$ under the distance metric $\Omega$. Formally, this assumption is expressed as

$$\forall \bar{x} \in \mathbb{R}^n, \; \Omega(x, \bar{x}) \leq \epsilon \implies f(\bar{x}) = f(x), \tag{4}$$

where $\Omega(x, \bar{x})$ quantifies the distance between $x$ and $\bar{x}$.

Consider a Trojaned input $\tilde{x} = x + v$, where $v$ denotes the injected trigger pattern. We assume that the Trojan perturbation satisfies $\Omega(x, \tilde{x}) \leq \tilde{\epsilon}$, and that $f(\tilde{x}) \neq f(x)$ due to the adversarial effect of the trigger. Let $\hat{x}$ denote a reconstructed version of the original input $x$ from the Trojaned input $\tilde{x}$, such that the reconstruction error is no greater than the perturbation itself, i.e., $\Omega(x, \hat{x}) \leq \Omega(x, \tilde{x})$. Under the condition that $\tilde{\epsilon} \leq \epsilon$, it satisfies

$$\Omega(x, \hat{x}) \leq \tilde{\epsilon} \leq \epsilon, \tag{5}$$

which implies that $\hat{x}$ lies within the $\epsilon$-ball centered at $x$. By the assumption in Eq. equation 4, it then holds that $f(\hat{x}) = f(x)$.

Therefore, if the reconstruction error is bounded by the perturbation magnitude $\tilde{\epsilon}$ and $\tilde{\epsilon} \leq \epsilon$, the reconstruction process yields a sample $\hat{x}$ that remains within the model's trusted prediction region. In this case, $\hat{x}$ preserves the original classification, effectively restoring the model's behavior on Trojaned inputs. $\square$

*Proof of Proposition 2.* Given a recovery image $\hat{x}$, the performance of attribution methods for a target class $c$ evaluated in our BackX benchmark can be represented as $I(\hat{x}; c)$. We follow the derivation on the multi-information as Vergara & Estévez (Vergara & Estévez, 2014), Rong et al. (Rong et al., 2022). Assume an attribution mask $S$, the multi-information $I(\hat{x}; c; M)$ can be represented as

$$I(c; \hat{x}|S) = I(c; \hat{x}|S) - I(\hat{x}; c), \tag{6}$$

$$I(\hat{x}; c; S) = I(c; S|\hat{x}) - I(S; c). \tag{7}$$

By equating Equation 6 and Equation 7, we can derive

$$I(\hat{x}; c) = I(c; \hat{x}|S) + I(S; c) - I(c; S|\hat{x}), \tag{8}$$

where $I(c; \hat{x}|S)$ represents the target we aim to estimate, representing the mutual information between the recovery input $\hat{x}$ and a class $c$ for a given mask $S$. $I(S; c)$ represents the mutual information between $S$ and $c$, which we aim to eliminate. The last term $I(c; S|\hat{x})$ indicates mutual information between $S$ and $c$ given $\hat{x}$, compensating for $I(S; c)$.

Minimizing the mutual information $I(S; c)$ between the mask and the class label leads to the approximation that $I(S; c)$ approaches $I(c; S|\hat{x})$, specifically $I(S; c) \approx I(c; S|\hat{x})$. Thus, this mitigation of the leaked information $I(S; c)$ from the mask $S$ leads to an improvement in the reliability of the evaluation results for $I(\hat{x}; c)$, specifically $I(\hat{x}; c) \approx I(c; \hat{x}|S)$. $\square$

## A.4 BENCHMARK FIDELITY DISCUSSION

In Tab. 1, we present a comparative analysis of different benchmarks with regard to their fidelity criteria assurance. Below, we discuss the details of the ratings provided in the table following three categories of the methods based on their underlying techniques.

**Input Perturbation-based Methods.** Since attribution maps are expected to highlight important input features, attribution benchmarks, including MoRF, LeRF (Samek et al., 2016), Ins.&Del. Games (Petsiuk et al., 2018), DiffID (Yang et al., 2023a) and ROAD (Rong et al., 2022), assess attribution methods by iteratively replacing pixels with zero or noise pixels. However, the input change causes an input distribution shift (Hooker et al., 2019), compromising the reliability of evaluation outcomes. To ensure *input distribution invariance*, ROAR (Hooker et al., 2019) and DiffROAR (Shah et al., 2021) retrain the model using perturbed input samples to incorporate the perturbed input samples within the training distribution. However, retraining models violate the criterion of *functional mapping invariance*. Thus, input perturbation-based attribution benchmarks shown in columns 1-3 of Tab. 1 struggle to ensure invariance to input distribution and the explained model.

**Sanity and Sensitivity Checks.** Sanity and sensitivity checks are originally proposed to test the fidelity of attribution methods for the explained input samples and models. Benchmarks such as sensitivity-n (Ancona et al., 2018), $SENS_{MAX}$, and INFD (Yeh et al., 2019) aim to assess the sensitivity of attribution methods under input perturbations. While these benchmarks provide valuable assessment targets, these targets are unachievable by attribution methods in practice, resulting only in a partial guarantee of *attribution verifiability*. Adebayo et al. (Adebayo et al., 2018) randomized model parameters and training labels to test the sanity of attributions. However, this sanity check can lead to a functional mapping shift. Similar to the input perturbation-based techniques, sanity and sensitivity checks in columns 4-5 of Tab. 1 rely on input or model perturbations, making it challenging to simultaneously satisfy both *input distribution invariance* and *functional mapping invariance*.

**Benchmark with Attribution Ground Truth.** Other attribution benchmarks have made efforts to establish attribution ground truth for assessing attribution methods. Pointing Game (Zhang et al., 2018), DiFull & DiPart (Rao et al., 2022) utilize training annotations as attribution ground truth. However, training annotations only offer partial ground truth information for attributions. Khakzar et al. (Khakzar et al., 2022) calculated Null Feature without class information through model optimization as the ground truth for attributions. However, there remains a notable absence of evidence substantiating the relationship between the features generated by model optimization and their corresponding attributions. CLEVR-XAI (Arras et al., 2022) employs a visual question-answering framework to generate synthetic images to provide controlled evaluations of generated objects. However, the presence of shadows from these objects compromises the strict confinement of attribution to the objects' physical areas. Lin et al. (Lin et al., 2021) employ neural Trojan for providing attribution ground truth, but their benchmark fails to guarantee that the model prediction change is solely attributable to Trojaned features. In addition to the partial guarantee of *attribution verifiability*, these attribution benchmarks evaluate the ratio between the defined ground truth region and the region with high attributions, compromising sensitivity in their evaluation metrics for attributions. From Tab. 1, it is evident that a full guarantee of attribution verifiability presents a significant challenge.

## A.5 FIDELITY EXAMINATION OF ATTRIBUTION VERIFIABILITY

To guarantee attribution verifiability, we carefully devise techniques for both backdoor and metric design to ensure that the prediction changes are fully attributed to the trigger features. During backdooring the model, we maintain low loss for both poisoned and clean samples while ensuring that trigger features substantially alter the prediction accuracy on poisoned samples, achieving up to 100% success as shown in Tabs. 3-5 of Supp. A.7.2.

Although the exclusivity of the trigger in the backdoored model helps the fidelity of the proposed metrics, logit and probability fractional change require a further careful design to ensure correct attribution ground truth. Consider the logit fractional change metric $\Delta \tilde{f}_y(\hat{x})$ defined in Eq. 3 that examines the extent to which the logit of the recovered sample $\hat{x}$ can be restored from a poisoned sample $\tilde{x}$ for the clean class $y$. It should be noted that the poisoned sample always contains class

information corresponding to the class $y$. Consequently, directly computing the fractional change in output logits, i.e., $\tilde{f}_y(\hat{x})/\tilde{f}_y(x)$, may compromise the reliability of benchmarking results, given the intrinsic information of $y$. To address this concern, we eliminate this potential class information by subtracting the predictive confidence $\tilde{f}_y(\tilde{x})$ on samples of the class $y$ from the metrics, thus ensuring a reliable assessment.

### A.6 ATTRIBUTION METHODS AND CATEGORIES

In our experiments, nine attribution methods are benchmarked including GCAM (Selvaraju et al., 2017), FullGrad (Srinivas & Fleuret, 2019), Grad (Simonyan et al., 2014), GGCAM (Selvaraju et al., 2017), SG (Smilkov et al., 2017), IG (Sundararajan et al., 2017), IG-Uniform (Sturmfels et al., 2020), AGI (Pan et al., 2021), and LPI (Yang et al., 2023a). We categorize these methods into three groups; namely *CAM-based*, *gradient-based*, and *integration-based methods*, based on their distinct underlying explanation processes, as explained below.

**CAM-based Methods.** Zhou et al. (Zhou et al., 2016) proposed a class activation mapping (CAM) technique to localize the class-specific features of input samples. Assuming a feature map $A^k$ of the $k$-th convolutional layer before a softmax layer, CAM-based attribution methods $M_{CAM}^c$ (e.g., GCAM) calculate a weighted combination of activation maps $A^k$ for a class $c$ as

$$M_{\text{CAM}}^c(x) = \Psi(ReLU(\sum_k w_k^c A^k)), \tag{9}$$

where $w_k^c = avg(\sum_i \sum_j \partial f_c(x)/\partial A_{i,j}^k)$ indicates the activation weight by applying a global average pooling on the activation map $A^k$, and $\Psi$ indicates an interpolation operator to estimate an attribution map with the same scale as $x$ from a down-sampled activation map. In our experiments, we test two CAM-based attribution methods including GCAM and FullGrad. In contrast to CAM, GCAM avoid the use of additional structures enabling a generalized applications. FullGrad further integrates gradients of model biases to ensure the completeness axiom(Sundararajan et al., 2017). Since FullGrad retains a similar process of utilizing an interpolated activation map, we categorize it alongside GCAM as a CAM-based method. Although other CAM-based attribution methods also exist (Jiang et al., 2021; Muhammad & Yeasin, 2020), we test two representative ones in our experiments to cover this category.

**Gradient-based Methods.** Gradient-based attribution methods, such as Grad (Simonyan et al., 2014), employ a first-order Taylor expansion to approximate the non-linear model: $f_c(x) \approx w^\intercal * x + b$. Thus, the attribution of an input sample $x$ for a class $c$, can be computed using the gradients with respect to the input as

$$M_{\text{Grad.}}^c(x) = \partial f_c(x)/\partial x. \tag{10}$$

In contrast with CAM-based attribution methods, gradient-based methods are capable of estimating attribution values at the element level for each input feature, eliminating the need for interpolation operators. In our experiment, we categorize GGCAM alongside Grad and SG as gradient-based attribution methods. The GGCAM leverages Grad to guide GCAM in performing element-wise attribution estimation.

**Integration-based Methods.** Compared to gradient-based attribution methods, integration-based attribution methods integrate input gradients from a reference input $x'$ to the feature $x$ along an integral path. Taking IG as an example, the attribution of an input feature $x_i$ for a class $c$ can be estimated as

$$M_{i\text{ Integr.}}^c(x, x') = (x_i - x_i') \times \int_{\alpha=0}^1 \left.\frac{\partial f_c(\tilde{x})}{\partial \tilde{x}_i}\right|_{\tilde{x}=x'+\alpha(x-x')} d\alpha, \tag{11}$$

where $\alpha$ indicates a linear integration path from the reference input $x'$ to the input $x$, and $x'$ is typically set as a zero vector in IG. Different integration-based attribution methods (IG-Uni, IG-SG (Smilkov et al., 2017), AGI, and LPI) are proposed to redefine the reference and the integral path. As these methods share the approach of estimating integrals to offer explanations, we categorize these methods as integration-based attribution methods. Due to their distinct theoretical guarantees in using different references (Sundararajan et al., 2017; Yang et al., 2023a), we benchmark four integration-based attribution methods including IG, IG-Uni, AGI and LPI.

Figure 9: Illustrations of input samples, trigger patterns and poisoned samples. The top arrow depicts fixed trigger patterns and corresponding poisoned samples with triggers in varying visibility ($\alpha$) used in the Blend attack. The bottom arrow illustrates invisible input-specific trigger patterns and corresponding poisoned samples generated using the ISSBA attack.

**Other Methods.** A variety of attribution methods also have been proposed for providing explanations through feature perturbation or removal (e.g., LIME (Ribeiro et al., 2016), SHAP (Lundberg & Lee, 2017), occlusion (Zeiler & Fergus, 2014), and mask (Fong & Vedaldi, 2017)). However, these perturbation-based methods are revealed to exhibit unreliabilities (Shrikumar et al., 2017; Sundararajan et al., 2017), as well as entailing a significant computational cost (Ancona et al., 2018). Therefore, we exclusively evaluate the propagation-based attribution methods. On the other hand, a few attribution methods, e.g., LRP (Binder et al., 2016) and DeepLift (Shrikumar et al., 2017), fail to satisfy basic axioms like *completeness* and *implementation invariance*. Hence, these methods are not included in our evaluation.

Overall, we benchmark two *CAM-based* methods (GCAM and FullGrad), three *gradient-based* methods (Grad, GGCAM and SG), and four *integration-based* methods (IG, IG-Uni, AGI and LPI).

### A.7 EXPERIMENTAL SETUP

In this section, we present a comprehensive experimental setup and hyperparameter choice for Trojaned models and benchmarked attribution methods, as well as details of the used experimental software and platform.

#### A.7.1 TRIGGER PATTERNS AND POISONED SAMPLES

**Related Works of Neural Trojans.** Backdoors alter a model's predictions by making it sensitive to certain trigger patterns. This framework is relevant because we leverage backdoors in our BackX. Gu et al. (Gu et al., 2019) introduced BadNet which mislabels and stamps triggers to poison the training samples. Blended attack (Chen et al., 2019), ISSBA (Li et al., 2021) and WaNet (Nguyen & Tran, 2021) are proposed to generate more stealthy sample-specific triggers, achieving invisible poisoning. Adap-Blend (Qi et al., 2022) further enhances the stealthiness of attack in the latent space. Due to the clear attribution of backdoor triggers in the models trained to achieve it, we leverage neural backdoors to circumvent the lack of ground truth in attribution evaluation. Given the wide applications of attribution methods in backdoor defense (Huang et al., 2019; Chou et al., 2020), we provide guidance for defense against backdoor attacks using attributions as part of this study.

**Neural Trojan Setup for BackX.** In our BackX benchmark, we incorporate both visible and invisible trigger patterns to provide a comprehensive evaluation, which allows us to thoroughly assess the capability of attribution methods in detecting both fixed visible patterns and input-specific invisible triggers. Figure 9 provides examples of the trigger patterns utilized in our experiments and the corresponding poisoned samples. In the Blend attack (Chen et al., 2019), we utilize fixed visible trigger patterns, consistent with Qi et al. (Qi et al., 2022). In addition, we employ a pre-trained encoder network to generate input-specific invisible trigger patterns in the ISSBA attack (Li et al., 2021). The trigger patterns used in these attacks are depicted in the middle column of Figure 9. Given a set of input samples, as shown in the first column of Figure 9, the resulting poisoned samples are presented in the last column of Figure 9.

Table 3: Comparison of accuracy among various ResNet-18 models Trojaned through Blend attack with triggers of varying visibility on CIFAR-10. The accuracy of the benign model (N/A) is also reported.

| Test set | Trigger visibility ($\alpha$) | | | | | | |
|---|---|---|---|---|---|---|---|
| | N/A | 0.1 | 0.3 | 0.5 | 0.7 | 0.9 | 1.0 |
| Poisoned | 0.008 | 0.993 | 0.999 | 1.000 | 1.000 | 1.000 | 1.000 |
| Clean | 0.943 | 0.936 | 0.933 | 0.928 | 0.935 | 0.937 | 0.933 |

Table 4: Comparison of accuracy among various ResNet-18 models Trojaned through Blend attack with triggers of varying visibility on GTSRB. The accuracy of the benign model (N/A) is also reported.

| Test set | Trigger visibility ($\alpha$) | | | | | | |
|---|---|---|---|---|---|---|---|
| | N/A | 0.1 | 0.3 | 0.5 | 0.7 | 0.9 | 1.0 |
| Poisoned | 0.001 | 0.992 | 0.999 | 1.000 | 1.000 | 1.000 | 1.000 |
| Clean | 0.973 | 0.967 | 0.965 | 0.970 | 0.971 | 0.967 | 0.969 |

Table 5: Comparison of accuracy among various ResNet-34 models Trojaned through Blend attack with triggers of varying visibility on ImageNet. The accuracy of the benign model (N/A) is also reported.

| Test set | Trigger visibility ($\alpha$)) | | | | | |
|---|---|---|---|---|---|---|
| | N/A | 0.3 | 0.5 | 0.7 | 0.9 | 1.0 |
| Poisoned | 0.000 | 0.987 | 0.998 | 0.995 | 0.999 | 1.000 |
| Clean | 0.724 | 0.713 | 0.717 | 0.715 | 0.714 | 0.716 |

### A.7.2 PERFORMANCE COMPARISON OF EXPLAINED MODELS

In this part, we show the detailed performance of used models trained on different datasets. For all the Trojaned models under comparison, we select the target label as the first class for a single target backdoor attack on both the CIFAR-10 and ImageNet datasets, and the third class for the GTSRB dataset followed by Qi et al. (Qi et al., 2022).

Table 6: Comparison of accuracy among various ResNet-18 models Trojaned through Blend attack with different poisoning rates on CIFAR-10.

| Test set | Poisoning rate | | | | |
|---|---|---|---|---|---|
| | 0.001 | 0.005 | 0.01 | 0.05 | 0.1 |
| Poisoned | 0.975 | 1.000 | 0.999 | 1.000 | 1.000 |
| Clean | 0.937 | 0.938 | 0.936 | 0.939 | 0.928 |

Table 7: Comparison of accuracy among various ResNet-18 models Trojaned through Blend attack with different poisoning rates on GTSRB.

| Test set | Poisoning rate | | | | |
|---|---|---|---|---|---|
| | 0.001 | 0.005 | 0.01 | 0.05 | 0.1 |
| Poisoned | 0.745 | 0.997 | 0.998 | 1.000 | 1.000 |
| Clean | 0.969 | 0.966 | 0.971 | 0.970 | 0.970 |

Tables 3 and 4 display the accuracy comparison of Trojaned ResNet-18 models as the visibility of the Trojan trigger varies on the CIFAR-10 and GTSRB datasets. Similarly, Table 5 compares the accuracy of different Trojaned ResNet-32 models on ImageNet. In all cases, a consistent poisoning rate of 0.1 was employed during model Trojaning. Notably, these tables illustrate that changes in trigger visibility do not lead to a marked decrease in accuracy on the clean dataset. Additionally,

Table 8: Comparison of accuracy among various ResNet-18 models Trojaned through ISSBA attack on CIFAR-10 and GTSRB.

| Test set | Dataset | |
|---|---|---|
| | CIFAR-10 | GTSRB |
| Poisoned | 1.000 | 1.000 |
| Clean | 0.938 | 0.972 |

Table 9: Accuracy comparison of ResNet-18, ResNet-32, ResNet-50 and ResNet-101 models Trojaned through Blend attack on ImageNet.

| Test set | Model | | | |
|---|---|---|---|---|
| | ResNet-18 | ResNet-34 | ResNet-50 | ResNet-101 |
| Poisoned | 0.998 | 0.998 | 0.993 | 0.990 |
| Clean | 0.682 | 0.717 | 0.728 | 0.749 |

there is a minor reduction in accuracy on the poisoned dataset to ensure a fair comparison. These results provide compelling evidence that a sample containing the Trojan trigger can completely alter the model's predictions, providing the ground truth of attributions. The first column in each table compares the accuracy of the benign model on the respective datasets, highlighting that Trojaned models only exhibit a slight performance compromise compared to their benign counterparts.

Table 6 and Table 7 present the accuracy changes in models Trojaned with different poisoning rates, ranging from 0.001 to 0.1, on the CIFAR-10 and GTSRB datasets. These tables demonstrate that models Trojaned with various poisoning rates only result in a subtle reduction in clean accuracy. It enables our experiments to maintain relatively consistent accuracy levels across different poisoning rates.

In Table 8, we show the accuracy comparison of ResNet models Trojaned through the ISSBA attack with a poisoning rate of 0.05 on both CIFAR-10 and GTSRB datasets. We have ensured consistent standard accuracy and high accuracy on the poisoned dataset to enable a fair comparison. Furthermore, Table 9 compares accuracy on both the poisoned and clean datasets across three different models.

The results highlight that Trojaned models exhibit the ability to adapt and fit the input distribution when optimizing the poisoned samples. As these Trojaned models converge during training on these samples, we are now equipped with the capacity to conduct a comprehensive and rigorous evaluation of attribution methods specifically within the context of Trojaned models. While other studies have demonstrated the presence of feature disparities in the hidden layers of Trojaned models within the latent space (Qi et al., 2022), our findings suggest that these models can still be considered reliable for evaluating explanations related to the model's output layer. However, further exploring the impact of the disparity in evaluating attribution methods is left in our future work.

### A.7.3 Hyperparameter Choice

*1) Model Training and Evaluation:* In our experiments, we trained ResNet-18 on CIFAR-10 for 100 epochs, starting with an initial learning rate of 0.1. We applied a learning rate decay by a factor of 10 at the 50-th and 75-th epochs separately. On GTSRB, ResNet-18 was trained for a total of 100 epochs, with a learning rate of 0.1, and we applied a learning rate decay at the 30-th and 60-th epochs. Additionally, we trained ResNet-18, ResNet-34, ResNet-50, and ResNet-101 on ImageNet 2012 training set for a total of 90 epochs, with an initial learning rate of $10^{-2}$, which was decayed at the 30-th and 60-th epochs. This setup was consistent for models subjected to both Blend attack and ISSBA attack. In the benchmarking, we use the test sets of both the CIFAR-10 and GTSRB datasets. For the ImageNet 2012 dataset, attribution methods are assessed on the ImageNet 2012 Validation set.

*2) Attribution Methods:* In our experiments, we test three groups of attribution methods including CAM-based, gradient-based and integration-based attribution methods. In two **CAM-based attribution methods**, GCAM and FullGrad, we remove the ReLU layer that is typically applied in

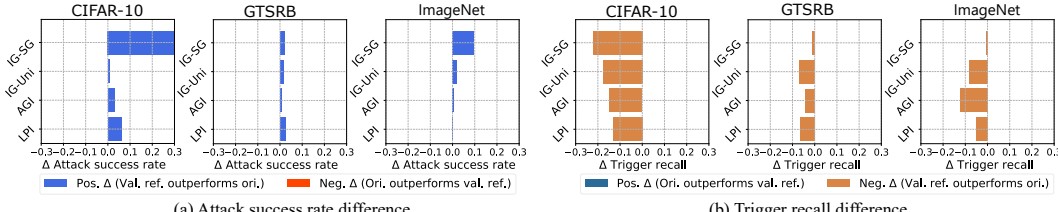

Figure 10: The comparison of **(a)** attack success rate difference and **(b)** trigger recall difference between original attributions and attributions recalibrated by valid references. Four integration-based attribution methods are compared to explain the model's output logit.

activation calculation, ensuring the original activations. In addition, we applied two CAM-based attributions to explain model output probabilities. For all **gradient-based attribution methods**, the attributions are calculated by explaining output logits. In addition, we take the absolute values of the calculated attributions of gradient-based methods. In SG, we integrate gradients of 50 perturbed input samples that are added Gaussian noise with a standard deviation of 0.15. In **integrated-based attribution methods**, we retain the original values of estimated attributions by explaining output probabilities. Specifically, we sample 50 interpolations $\bar{x}$ from the reference $x'$ to the input $x$ in IG for attribution estimation. In IG-Uni, IG-SG and LPI, we employ 10 references and 5 interpolations to maintain the same total number of interpolations of 50 for a fair comparison. Specifically, IG-SG employs the same deviation in Gaussian noise as SG. References of LPI are sampled from the training set by one central clustering. In contrast, we select 5 references and 10 interpolations in AGI for a higher performance due to its reliance on more interpolation points for estimating adversarial examples.

### A.7.4 EXPERIMENTAL SOFTWARE AND PLATFORM

All experiments were performed on a Linux-based system equipped with an NVIDIA GTX 3090Ti GPU boasting 24GB of memory, complemented by a 16-core 3.9GHz Intel Core i9-12900K CPU and 128GB of main memory. For both testing and training purposes, all attribution methods were implemented and evaluated within the PyTorch deep learning framework (version 1.12.1), utilizing the Python programming language.

### A.8 RE-CALIBRATED ATTRIBUTION AND CONTRASTIVE OUTPUT

Recent research has explored alternatives to enhance reliability of baseline attribution results (Wang & Wang, 2022), (Yang et al., 2023b). Here, we conduct comparative experiments for such choices. Re-calibrated attributions (Wang & Wang, 2022) have been introduced to enhance integration-based methods by estimating attributions from identified valid references. In Fig. 10, we compare the re-calibrated attributions and original attributions across integration-based attribution methods, including IG-SG, IG-Uni, AGI and LPI. It is observed that re-calibrated attributions further enhance the performance of the original attributions on CIFAR-10, GTSRB and ImageNet datasets. In addition, the performance of re-calibrated attributions is observed to degrade with decreasing mean input pixel values, aligning with our Observation 2. This demonstrates the sensitivity of integration-based methods to variations in the reference values. Despite the enhancement achieved through re-calibration, it remains unique to integration-based attribution methods. To maintain evaluation consistency across different methods, we retain the use of original attributions in the integration methods for the subsequent experiments.

Figure 11 offers a comparison between attributions computed by our established consistent setup and contrastive outputs (Wang & Wang, 2022) on CIFAR-10, GTSRB, and ImageNet. It is notable that explaining contrastive output yields similar performance compared to explaining manually selected outputs, particularly evident in cases involving FullGrad and integration-based methods. This observation underscores the superiority of contrastive output over mutual selection. However, explaining contrastive output in CAM and Gradient-based methods does not achieve comparable performance. The following observation is derived from the above analysis.

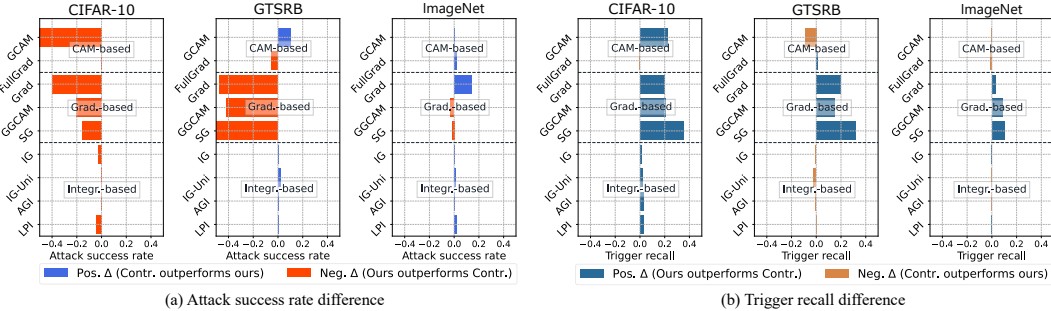

(a) Attack success rate difference    (b) Trigger recall difference

Figure 11: The comparison of **(a)** attack success rate difference and **(b)** trigger recall difference between attributions estimated from our established consistent setup and attributions calculated by explaining the contrastive output.

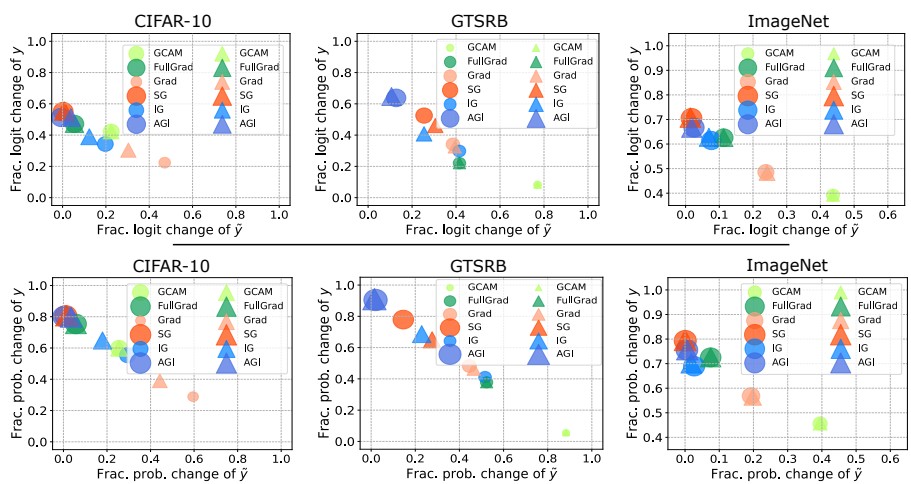

Figure 12: Comparison of fractional logit change (**top row**) and fractional probability change (**bottom row**) between explaining logits and probabilities. Approaching top-left corner in each plot indicates superior performance. ● denotes that probabilities are explained with attribution methods. ▲ indicates that logits are explained with the attribution methods.

**Observation 3.1** *The re-calibration technique consistently improves the performance of integration-based attribution methods. Attribution derived from explaining contrastive model output can achieve competitive results, particularly for CAM-based and integration-based methods.*

## A.9 EXTENDED EXPERIMENTS OF OUTPUT CHOICE

In this part, we provide more experimental results of attribution methods in explaining different objects.

Let $p(x) = \text{softmax}(\tilde{f}(x))$ denote the probability output through a softmax function. Similar to the logit fractional change defined in Equation 3, we define the fractional change of output probabilities $\Delta p_{\tilde{y}}(\hat{x})$ and $\Delta p_y(\hat{x})$ can be defined to measure changes of normalized outputs in class $\tilde{y}$ and $y$.

In Fig. 12, we show the comparison of fractional logit and fractional probability change between the attributions across three datasets. This evaluation combines fractional change of both $y$ and $\tilde{y}$ to compare the capability of attribution methods in reducing output of $\tilde{y}$ and recovering output of $y$. All methods are scaled by their distance to the bottom left corner. Methods are represented by markers of a circle (●) and a triangle (▲) to separate attributions estimated in explaining output probabilities and logits. The evaluation allows us to assess whether the results of attribution methods change when the evaluation target is altered. It is observed that the performance of various attribution methods remains consistent when evaluating fractional change in output probabilities and logits

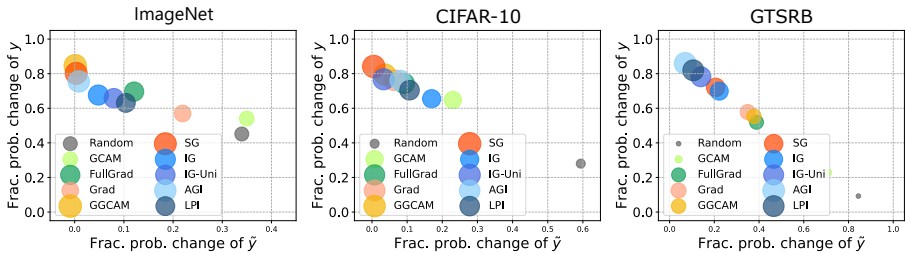

Figure 13: The comparison of fractional probability change under Blend attack. Different attribution methods are scaled according to their performance. Approaching top-left corner in each plot indicates superior performance.

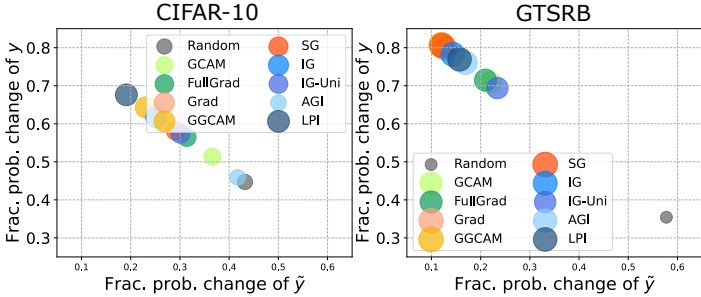

Figure 14: The comparison of fractional probability change under ISSBA attack. Different attribution methods are scaled according to their performance. Approaching top-left corner in each plot indicates superior performance.

within a specific dataset. This underscores the consistency in the preferences of attribution methods across different levels of predictive confidence, regardless of the specific objective being explained. These findings indicate the crucial role of establishing a clear and well-defined explanation setup in attribution methods.

### A.10 EXTENDED EXPERIMENTS OF OVERALL BENCHMARKS

In this part, we provide more overall benchmarking experimental results. In addition, more visualization examples are presented for a visual inspection.

#### A.10.1 RESULTS OF FRACTIONAL PROBABILITY CHANGE

Figure 13 and Figure 14 presents a comparison of fractional probability change across attribution methods for both Blend and ISSBA attacks. It is evident that gradient-based and integration-based attribution methods consistently exhibit the highest performance and stability in reducing the output of $\tilde{y}$ and recovering the output of $y$. CAM-based methods, including GGCAM, also demonstrate high performance, relying on a combination of fine-grained attributions.

#### A.10.2 RESULTS OF OTHER ATTRIBUTION METHODS

In this study, we primarily aim to enhance model reliability, which led us to focus on attribution methods with established robustness. Consequently, SHAP-based and perturbation-based approaches were excluded from the main benchmark, as previous works have highlighted their reliability limitations (Shrikumar et al., 2017; Sundararajan et al., 2017). Furthermore, perturbation-based methods are generally more computationally intensive compared to propagation-based alternatives. However, to assess their potential contributions, we extend our evaluation to include additional attribution methods that were not the primary focus of our study. These methods include LIME (Ribeiro et al., 2016), LRP (Binder et al., 2016), DeepLIFT (Shrikumar et al., 2017), KernelSHAP (Lundberg & Lee, 2017), GradientSHAP (Lundberg & Lee, 2017), HSIC (Novello et al., 2022), and SMDL-Shap (Chen et al., 2024), alongside IG, SG, and FG, using ImageNet on MobileNet-V2.

Figures 15(a) and 15(b) present a comparison of the attack success rate and trigger recall for both focused and unfocused attribution methods. The results indicate that the unfocused methods tend to perform less favorably than the focused methods. Additionally, we observe that SHAP-based methods exhibit high sensitivity to the choice of feature set, further complicating their application.

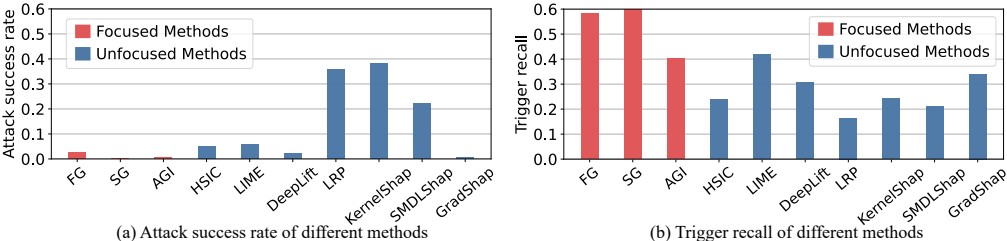

(a) Attack success rate of different methods    (b) Trigger recall of different methods

Figure 15: Comparative analysis of focused and unfocused attribution methods using **(a)** attack success rate and **(b)** trigger recall on MobileNetV2 trained on ImageNet. A **lower** attack success rate signifies **better** performance, while a **higher** trigger recall indicates **better** performance.

### A.10.3 EXTENDED EXPERIMENTS ON MODEL ARCHITECTURES AND DEPTHS

In this part, we examine the impact of different model architectures and model depths on attribution performance.

*1) Different Model Architectures:* Figure 16 compares the attack success rate and trigger recall across various model architectures, including MobileNetV2 (Sandler et al., 2018), VGG16 (Simonyan & Zisserman, 2015), and ResNet-50 (He et al., 2016), using different attribution methods. All models were trained on ImageNet with a Trojan trigger of 0.5 visibility and a 0.1 poisoning rate. The results show that two CAM-based attribution methods and input gradient attribution methods are sensitive to changes in model architecture, demonstrating lower stability. In contrast, other evaluated attribution methods exhibit consistent performance across different architectures, suggesting higher stability. Additionally, ResNet-50, when compared to MobileNetV2 and VGG16, proves to be more challenging to explain, as evidenced by its higher attack success rate and lower trigger recall.

*2) Different Model Depth:* Figure 17 compares attribution methods on ResNet models of varying depths, assessing both attack success rate and trigger recall on ImageNet. Similarly, the two CAM-based methods and input gradient attributions show instability when explaining models of different depths. On the other hand, the other attribution methods exhibit consistent performance across models of varying depths. Overall, model depth does not significantly complicate the task of attribution.

Based on these evaluations, we make the following observations:

**Observation 5.** *CAM-based and vanilla input gradient attributions are sensitive to model changes. In contrast, most other attribution methods show consistent performance across different models. Overall, more complex architectures generally increase the difficulty of attribution, whereas changes in model depth pose less of a challenge.*

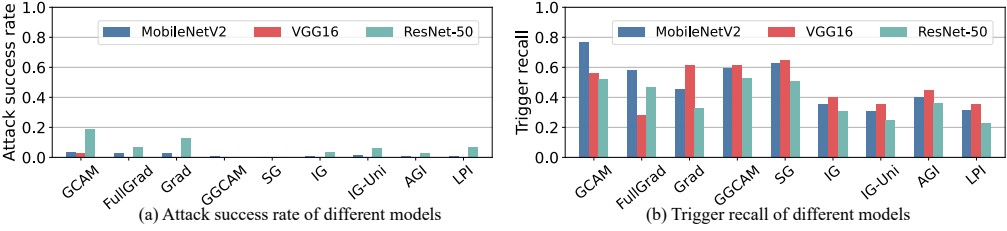

(a) Attack success rate of different models    (b) Trigger recall of different models

Figure 16: The comparison of different attribution methods using **(a)** attack success rate and **(b)** trigger recall on different model architectures trained on ImageNet. A **lower** attack success rate indicates **better** results, and a **higher** trigger recall indicates **better** results.

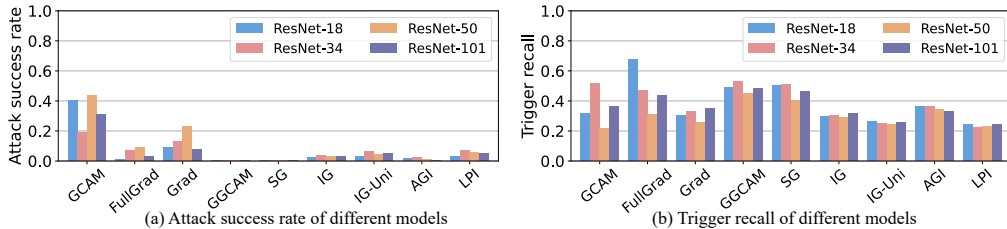

Figure 17: The comparison of different attribution methods using **(a)** attack success rate and **(b)** trigger recall on ResNet of depth 18, 34, 50, and 101 trained on ImageNet. A **lower** attack success rate indicates **better** results, and a **higher** trigger recall indicates **better** results.

Table 10: Accuracy comparison of Bert-based models Trojaned through BadNet attack on SST-2, IMDB and Amazon Review datasets. The accuracy on both poisoned and clean testing set are presented.

| Test set | Sentiment Analysis Datasets | | |
|---|---|---|---|
| | SST-2 | IMDB | Amazon |
| Poisoned | 0.924 | 0.935 | 0.961 |
| Clean | 1.000 | 0.938 | 0.982 |

## A.11 EXTENDED BENCHMARKING RESULTS ON TEXTUAL ANALYSIS

Most attribution methods have been primarily focused on the image domain. In this work, we extend our benchmark to the textual domain, further demonstrating the applicability of BackX. In this section, we first outline the benchmarking setup and then provide a comprehensive evaluation of attribution methods in the context of textual analysis.

### A.11.1 TEXTUAL BENCHMARKING PROCESS

To benchmark attribution methods in the textual domain, we construct Trojaned language models using the BadNet attack (Gu et al., 2019), as implemented by OpenBackdoor (Cui et al., 2022). We consider four different trigger patterns–"*cf*", "*mn*", "*bb*", and "*tq*"-which are randomly injected at predefined locations within the input text while modifying the corresponding training labels to generate poisoned samples. We fine-tune a BERT-based model on three poisoned sentiment analysis datasets: SST-2 (Socher et al., 2013), IMDB (Maas et al., 2011), and Amazon Reviews (He & McAuley, 2016). Specifically, we employ a pre-trained BERT-based model (Kenton & Toutanova, 2019) with 12 transformer layers, 768 hidden units, and 12 attention heads as the victim model for textual Trojaning. Table 10 reports the performance of the Trojaned BERT models across the three datasets. The results indicate that neural Trojans cause significant alterations in the model's predictions, thereby confirming the fidelity of the subsequent benchmarking analysis.

For the benchmarked attribution methods, we exclude class activation maps (CAM) typically used in image classification tasks to highlight discriminative regions, as such maps are not directly applicable to textual classification. Instead, we focus on gradient-based and integral-based attribution methods, which are better suited for generating attributions in the textual domain. Unlike image-based models, where the input data is manipulated at the pixel level, textual inputs are first tokenized into discrete token IDs. This tokenization results in a representation that is not differentiable, as the input IDs are discrete and do not support gradient-based manipulation. Consequently, direct attribution to the raw input text is not feasible. To overcome this limitation, we compute attributions for the input features after the embedding layer, where the token IDs are transformed into continuous and differentiable representations. Thus, in our BackX benchmarking, we estimate attributions based on the embeddings of the input text.

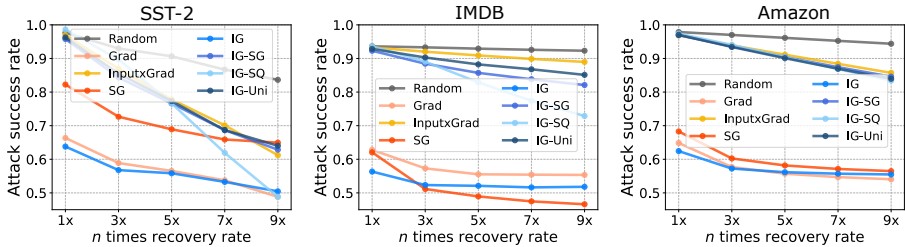

Figure 18: Benchmarking of attribution methods under BadNet attack using the attack success rate metric on Bert-based models trained on SST-2, IMDB and Amazon datasets. Lower curves indicate better results.

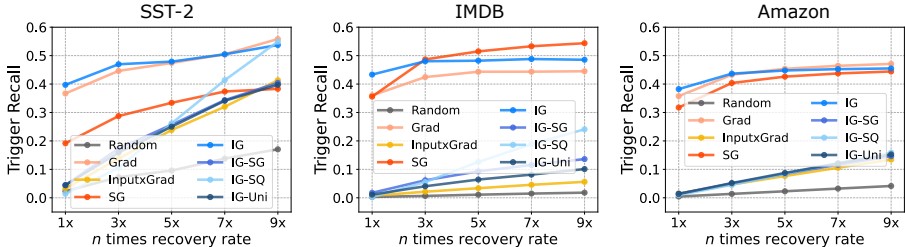

Figure 19: Benchmarking of attribution methods under BadNet attack using the trigger recall metric. Higher curves indicate better results.

### A.11.2 EVALUATION ON TEXTUAL ANALYSIS

Figures 18 and 19 present a comparison of various attribution methods using two evaluation metrics: attack success rate and trigger recall, across the SST-2, IMDB, and Amazon datasets. The recovery rate represents the proportion of tokens restored to their clean state from clean samples, where a recovery rate of $1\times$ corresponds to the number of tokens contained in the textual trigger pattern.

The results indicate that Integrated Gradients (IG), InputGrad (Grad), and SmoothGrad (SG) exhibit the most consistent and robust performance across all three datasets. Notably, IG's performance in the textual domain differs significantly from that in the image domain, underscoring its high reliability for generating explanations. This enhancement in reliability is attributed to the more reliable selection of reference inputs by IG. Specifically, in contrast to the vision domain, IG's use of zero-valued vectors as reference inputs accurately captures the absence of features in the textual domain. This distinction is reflected in the performance gap between IG and its variants, which employ alternative reference inputs such as Gaussian noise, uniform noise, and squared Gaussian noise. Moreover, due to the uncertainty introduced by perturbations transferring from the input to the feature space, IG's advantage becomes more pronounced when attributing features in the embedding layer, leading to superior performance.

Similarly, Grad demonstrates competitive results compared to SG, showing substantial improvements in the textual domain. SG's reduced effectiveness in text attribution can be attributed to its reliance on noisy gradients, which struggle to capture complex interactions in feature space. Due to the lower perturbation amplitude, its performance remains close to that of Grad. These findings suggest that perturbation-based attribution methods fail to enhance explanations in the textual domain effectively. Moreover, the observed variability in attribution reliability across domains underscores the necessity of tailoring attribution methods to improve explanation faithfulness. The benchmarking results on textual analysis further highlight the potential for extending our framework beyond textual data to other modalities, such as audio, time-series, and graphs.

In Fig. 20, we present visualization results of the attributions used to explain sentiment prediction outcomes on SST-2 dataset. For a given input sample, the poisoned sample containing the "*mn*" trigger causes a shift in the sentiment prediction from negative to positive. IG and Grad effectively attribute this prediction shift to the trigger pattern, whereas other attribution methods exhibit compromised performance in explaining this shift.

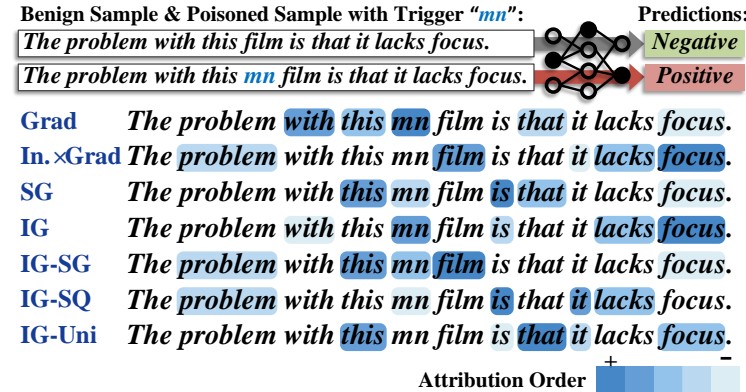

Figure 20: Attribution visualization for explaining a text sample from the SST-2 Dataset. Given an input sample, the poisoned sample using trigger "mn" alters the sentiment prediction from negative to positive. The importance of each input word, as estimated by different attribution methods, is visualized. Brighter colors indicate more important words.

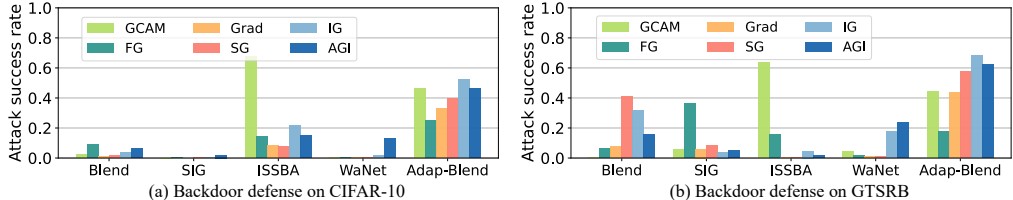

Figure 21: Comparative analysis of defending neural backdoors using different attribution methods on MobileNet-V2 model trained on **(a)** CFAIR-10 and **(b)** GTSRB datasets. We using attributions. For each poisoned sample, pixels corresponding to the top 50% highest attribution scores are restored to their clean state. **Lower** attack success rate indicates **better** defense performance.

**Observation 6.** *For textual analysis, IG and Grad achieve superior performance. IG, leveraging a zero reference, effectively mitigates challenges associated with reference selection by accurately capturing the absence of features. In contrast, perturbation-based attribution methods suffer degradation due to uncertainty in the feature space. The observed variability in attribution reliability across domains highlights the need for tailored attribution methods that account for domain-specific characteristics.*

## A.12 EXTENDED EXPERIMENTS OF BACKDOOR DEFENSE

In this part, more experiments are provided for defending against backdoor attacks using attribution methods. In addition, more visualization examples for detecting various triggers are provided for visual inspection.

### A.12.1 EXPLAINABILITY UNDER DIVERSE ATTACK METHODS

We investigate the effectiveness of attribution methods in identifying and mitigating various types of neural Trojans. Beyond the Blend and ISSBA attacks, we extend our analysis to additional backdoor methods, including SIG (Barni et al., 2019), WaNet (Nguyen & Tran, 2021), and Adap-Blend (Qi et al., 2022). SIG utilizes sinusoidal signals as triggers, seamlessly blending into textures to evade spatial filtering defenses. WaNet introduces geometric warping rather than pixel-level modifications, rendering backdoors visually imperceptible. Adap-Blend enhances stealth by reducing feature separability in the latent space, making Trojaned features less distinguishable from normal ones.

Figure 21 presents a comparative evaluation of attribution methods on MobileNet-V2 trained on CIFAR-10 and GTSRB. In this evaluation, we recover 50% of the poisoned sample pixels based on the attribution scores, restoring these pixels to their clean-state counterparts from unaltered samples.

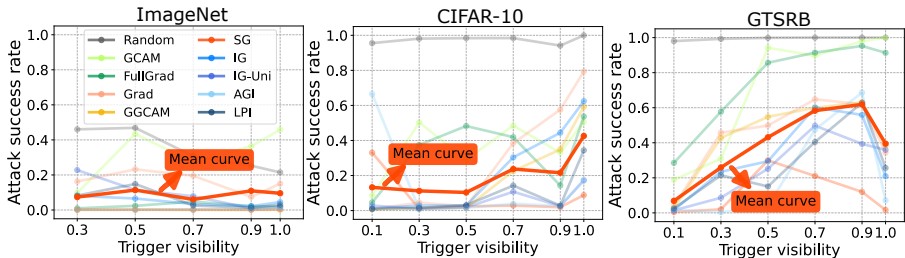

Figure 22: The results of attack success rate. Trigger recall changes as trigger visibility. Mean curves over various methods are highlighted. Lower values of attack success rate correspond to an increased capacity to defend against backdoor attacks.

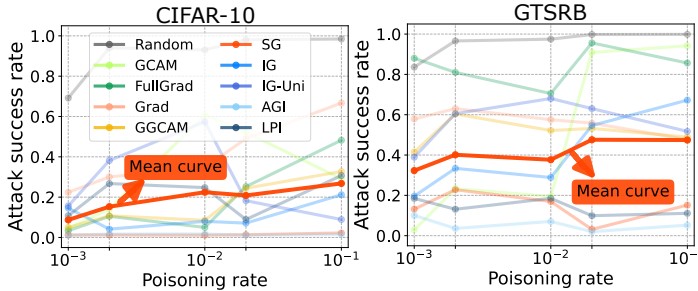

Figure 23: The results of attack success rate. Trigger recall changes as poisoning rate. Mean curves over various methods are highlighted. Lower values of attack success rate correspond to an increased capacity to defend against backdoor attacks.

The results indicate that CAM-based attribution methods are more effective in detecting triggers with fixed and consistent spatial patterns, as evidenced by their superior performance on Blend, Adap-Blend, and WaNet. In contrast, gradient-based and integral-based attribution methods excel in identifying discrete and imperceptible triggers, such as those used in SIG and ISSBA attacks.

However, attribution methods struggle to effectively neutralize the Adap-Blend backdoor. This limitation arises because Adap-Blend utilizes image-wide trigger patterns and applies regularization that encourages individual features affected by the trigger to independently influence the model's predictions. Consequently, even with a recovery rate of 50%, Adap-Blend maintains a relatively high attack success rate. These findings highlight the necessity of designing tailored attribution methods that account for the distinct characteristics of different backdoor strategies.

**Observation 7.** *CAM-based methods excel at detecting fixed spatial triggers, while gradient- and integral-based methods are more effective against discrete and imperceptible ones, highlighting the need for tailored attribution techniques for backdoor defending.*

### A.12.2 Results of Attack Success Rate

In Figure 22 and Figure 23, we compare the results of attack success rate on models Trojaned through Blend attack with different trigger visibilities and poisoning rates. It can be observed that the visibility of the trigger is not positively related to attribution performance. In contrast, the higher visibility trigger sometimes causes a higher attack success rate, which aligns with our observation. In addition, different poisoning rates show less impact on the ease of defense, as shown in Figure 23.

### A.13 Discussion and Limitation

In this section, we discuss the position of BackX in attribution evaluation and clarify its contribution to the fidelity benchmark alongside limitations.

*What number and diversity of Trojans are sufficient for BackX to deliver confident benchmarking results of attributions?* Similar to all existing attribution benchmarks, our underlying goal is to pro-

vide a necessary guarantee for rejecting unreliable attribution methods. In this context, any number of Trojans can provide sufficient confidence in evaluating attribution reliability. From the perspective of **necessary** conditions, since the fundamental requirement of any explanation method is to remain faithful to the model's prediction, all Trojans introduced in our framework are sufficient to provide strong evidence against unreliable attribution methods. If an attribution method fails to provide faithful explanations under any controlled Trojaning scenario, it should be rejected from real-world deployment. Compared to prior benchmarks (e.g., CLEVR-XAI, Null Feature, and DiPart), our method offers a fully controllable setting to test explanation reliability and further strengthens this ability by providing a tighter theoretical guarantee, thus offering a more solid reason for rejecting unreliable attribution methods, especially in high-stakes applications. From the perspective of **sufficient** conditions, while arguably no benchmark can guarantee that an attribution method is universally reliable, we strive to ascertain the sufficiency of BackX across diverse domains by extending beyond simple patch-based triggers with diverse trigger patterns, multiple fidelity metrics, and different modalities, thereby benchmarking attribution methods under varied and increasingly challenging conditions. This progressive expansion helps toward the sufficiency of our evaluation, while we acknowledge that sufficiency remains inherently limited across all existing benchmarks and explicitly aim to push this boundary further. In summary, our method offers a stronger reliability guarantee than existing benchmarks by enforcing a necessary condition for attribution methods. If an attribution method fails under our controlled setting, it provides a clear reason to reject its reliability. This necessity guarantee represents a central advantage of our approach over prior works.

*Does the mechanism of neural Trojans provide attribution fidelity?* Unlike prior works that detect backdoor triggers using attribution methods (Lin et al., 2021; Li et al., 2023), BackX is a high-fidelity benchmark established within rigorous fidelity criteria through tailored strategies with tighter theoretical guarantees. Therefore, it is not merely the existence of neural Trojans, but BackX's designs enable Trojans to serve as a pathway for achieving a reliable high-fidelity attribution benchmarking. Specifically, our framework formalizes fidelity as verifiable criteria and provides stronger guarantees by mitigating leakage and enforcing restoration so that prediction shifts are causally attributable. With such fidelity guarantees, BackX introduces a standardized evaluation setup across attribution paradigms, reducing confounding factors from post-processing and output forms. It also extends coverage substantially, testing fifteen attribution methods under five Trojan families across both vision and text. This comprehensive setup allows us to draw broader and more reliable conclusions. In Table 11, we report pairwise Kendall's $\tau$ between benchmark rankings for the seven attribution methods SG, AGI, IG, IG-SG, FG, IxG, and GCAM. The results show that BackX is broadly consistent with DiFull, DiffID, and ROAR, whereas NeuronTrojans aligns weakly with BackX and also exhibits reduced consistency with the other benchmarks. The results provide compelling evidence that a backdoored model on its own does not deliver benchmarking fidelity. Fidelity arises only when backdoors are embedded within a framework that enforces formal criteria, theoretically grounded procedures, and a standardized evaluation protocol. BackX treats backdoors as instruments to isolate attribution effects and to satisfy fidelity requirements, rather than as mere objects of detection.

Table 11: Pairwise Kendall's $\tau$ between benchmark rankings on seven attribution methods. Higher values indicate stronger agreement in method rankings.

| Benchmarks | Kendall's $\tau$ |
|---|---|
| BackX vs DiFull | 0.333 |
| BackX vs DiffID | 0.619 |
| BackX vs DiffROAR | 0.714 |
| BackX vs NeuronTrojans | 0.048 |
| NeuronTrojans vs DiFull | -0.238 |
| NeuronTrojans vs DiffID | 0.429 |
| NeuronTrojans vs DiffROAR | 0.143 |

*Limitation.* While BackX evaluates attribution methods under a diverse set of trigger patterns, it does not explicitly measure the stability of their performance across these variations. Moreover, although BackX offers a rigorous and necessary test for rejecting unreliable attribution methods, strong performance under its controlled settings may not necessarily imply full reliability in complex, real-world scenarios. Expanding BackX to better capture complex real-world attribution patterns and to generalize across broader domains remains an important direction for future research.

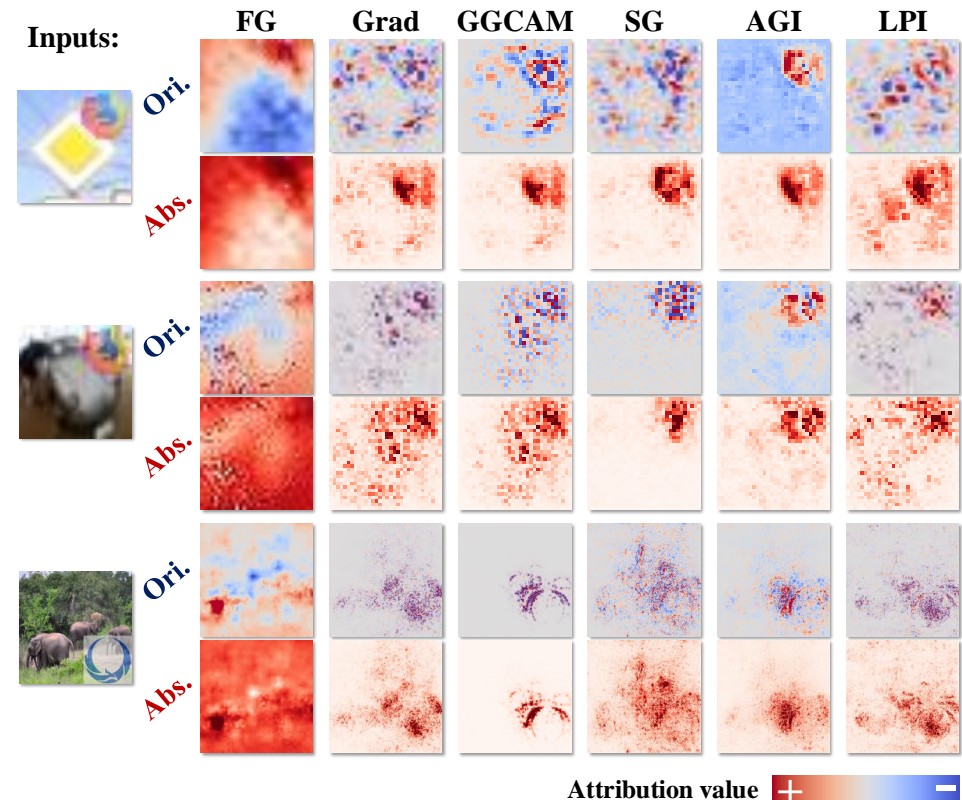

Figure 24: Visualization of absolute and original attributions for input samples from GTSRB, CIFAR-10 and ImageNet respectively. The predictions made by models backdoored through Blend attack with trigger visibility of $0.5$ are explained using attribution methods.

### A.14 VISUAL INSPECTION OF ATTRIBUTIONS

In Figure 24, we provide visualizations of employing absolute and original attributions for different methods. There is a notable disparity in the highlighted regions of input samples between the two post-processing techniques. This underscores the susceptibility of visual outcomes to ambiguity, emphasizing the necessity of quantitative assessments to provide clarity and context. This problem is more pronounced for smaller input images due to the limited number of pixels. Although large ImageNet size images may yield more consistent attribution maps, it is crucial to recognize that their relative feature importances undergo shifts, often reflected in attributions alternating between negative and positive values. This begs for an enquiry into the right choice between the original and absolute values for benchmarking.

In Figure 25 and Figure 26, we provide visualization examples for explaining models Trojaned through Blend attack using attribution methods across three datasets: CIFAR-10, GTSRB and ImageNet. Three groups of attribution methods are compared including CAM-based, Gradient-based and Integration-based methods. In the visualization experiments, original attributions of output logits are used in CAM-based and integration-based attribution methods. In addition, absolute attributions of output probabilities are visualized in gradient-based attribution methods as the established consistent setup. It can be observed that absolute attributions are good at generating plausible visualization examples. Similar to our observation, GCAM relies on other information to generate fine-grained attributions (e.g. FG and GGCAM).

Figure 27 presents visualization examples for explaining ResNet-18 models Trojaned through ISSBA attack using attribution methods of three groups on both CIFAR-10 and GTSRB datasets. We can observe that attributions with high variations are also effective in identifying invisible patterns in comparison with fixed patterns.

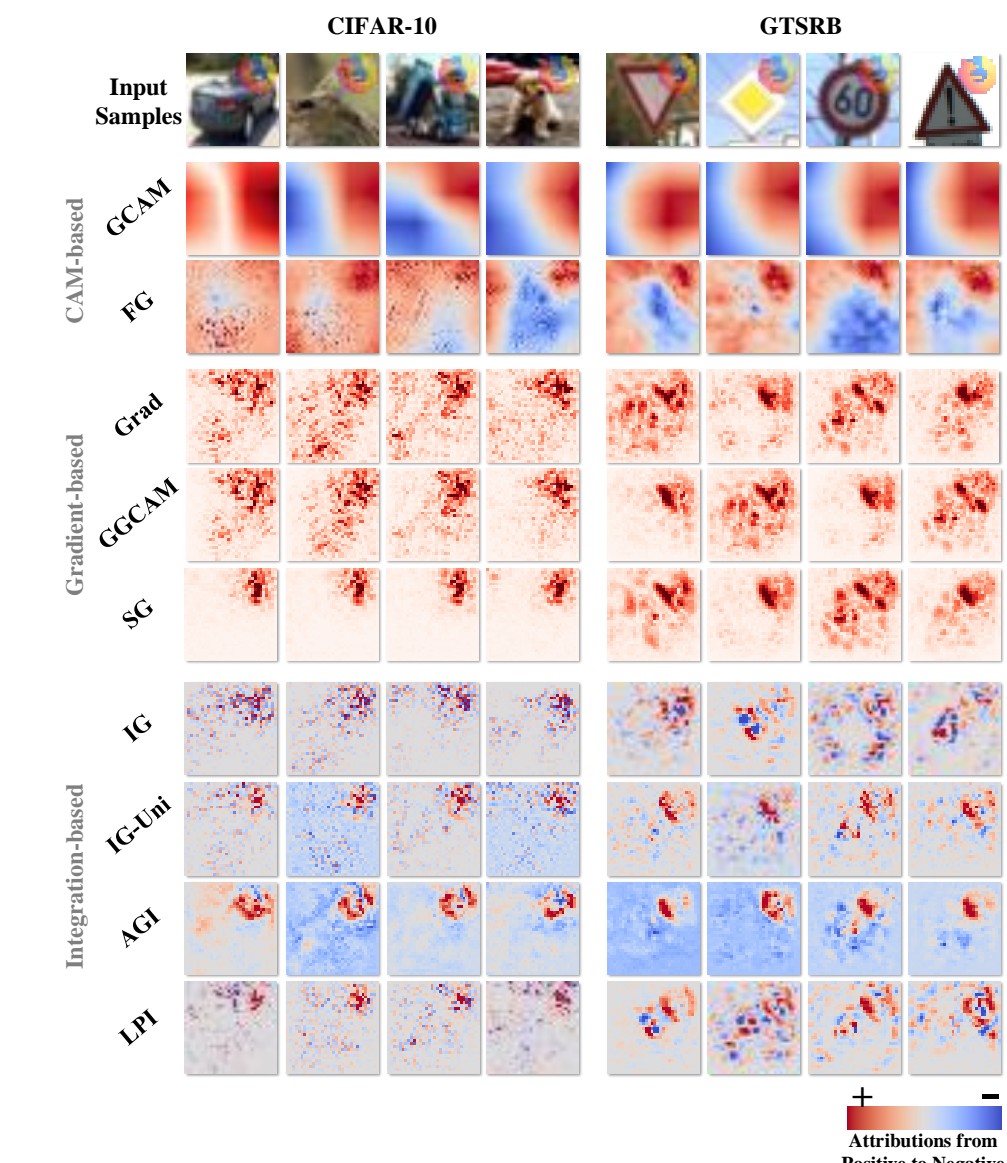

Figure 25: Visual comparison of attribution methods on CIFAR-10 and GTSRB datasets. Predictions made by ResNet-18 models Trojaned through the Blend attack with a trigger visibility of 0.5 are explained using three groups of attribution methods including CAM-based, Gradient-based, and Integration-based methods.

### A.14.1 VISUAL INSPECTION FOR BACKDOOR ATTACK

In Figure 28, we undertake an evaluation of attribution performance concerning the detection of triggers with varying degrees of visibility. Contrary to our intuitive assumptions, we observe that attribution methods encounter increased difficulty in identifying triggers with higher visibility, a trend that aligns with our empirical findings. This discovery prompts us to reconsider our intuition about feature learning within model optimization processes.

Figure 29 presents visualization examples of attribution maps with changes in the poisoning rate. Notably, it is apparent that attributions exhibit a relatively minimal influence on models subjected to different poisoning rates. This observation underscores the resilience of model behavior to alterations induced by varying levels of poisoning, highlighting the intricate dynamics at play in adversarial model manipulation.

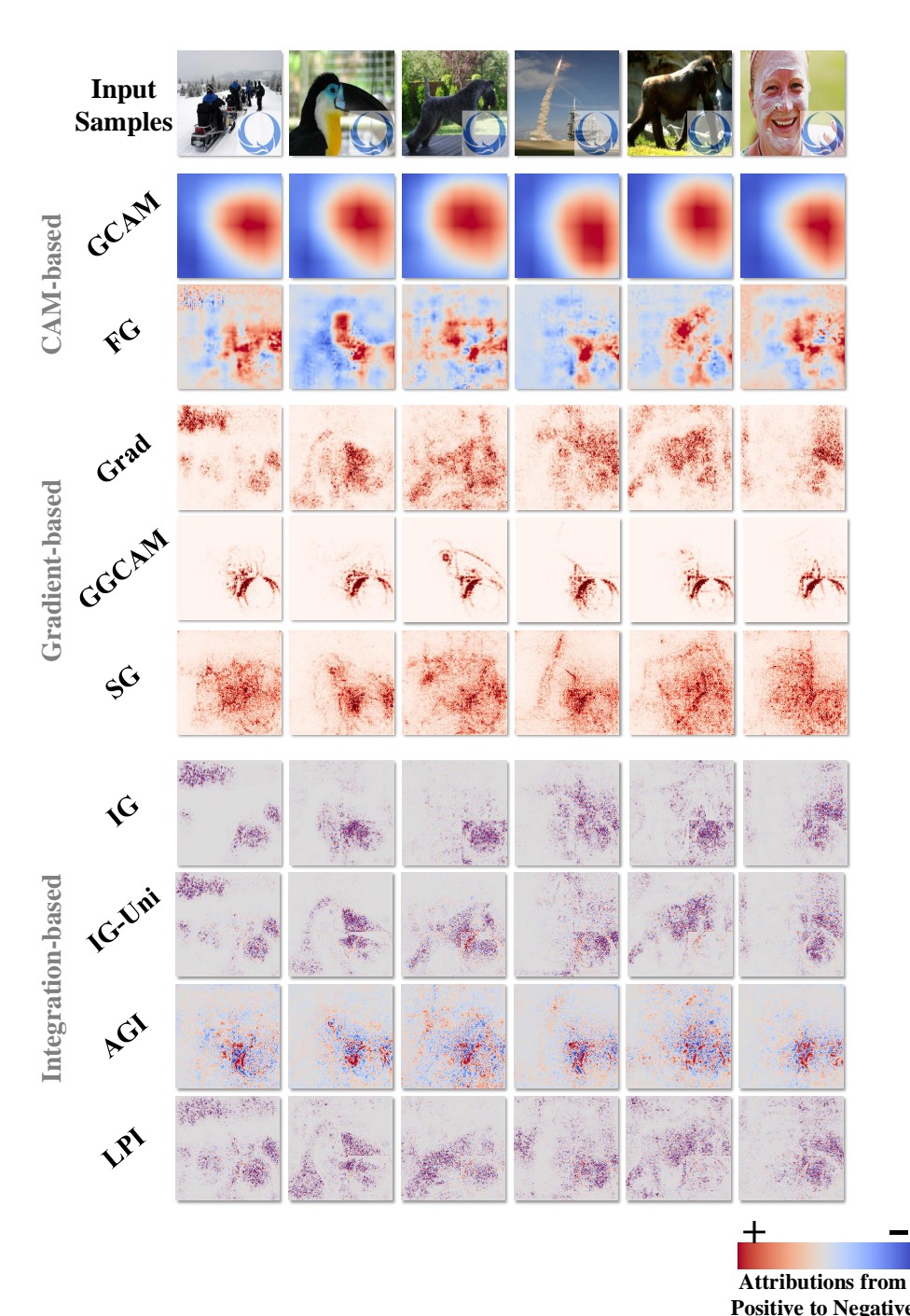

Figure 26: Visual comparison of attribution methods on ImageNet 2012. Predictions made by ResNet-34 models Trojaned through the Blend attack with a trigger visibility of 0.5 are explained using three groups of attribution methods including CAM-based, Gradient-based, and Integration-based methods.

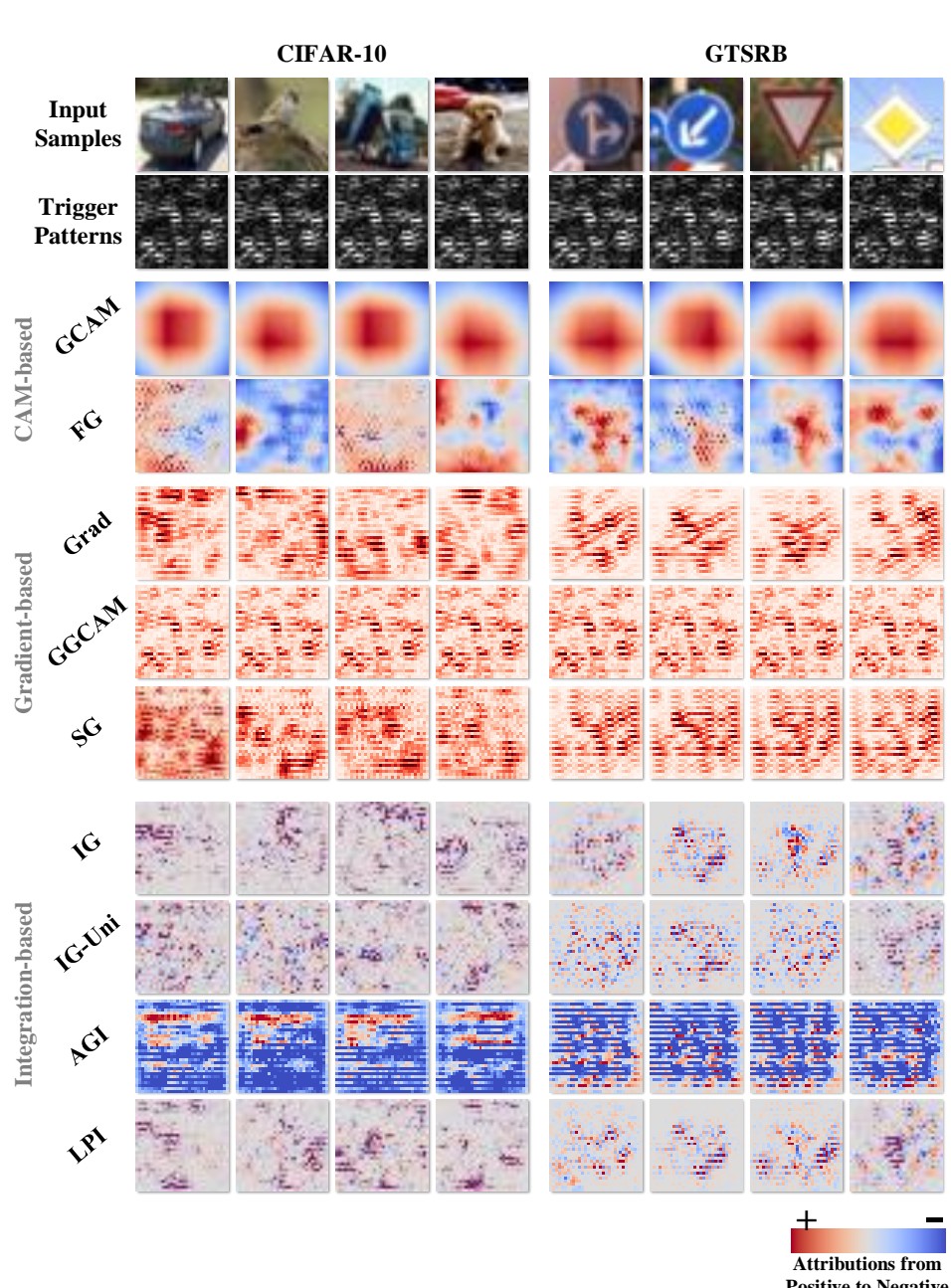

Figure 27: Visual comparison of attribution methods on CIFAR-10 and GTSRB datasets. Predictions made by ResNet-18 models Trojaned through the ISSBA attack with a trigger visibility of $0.5$ are explained using three groups of attribution methods including CAM-based, Gradient-based, and Integration-based methods.

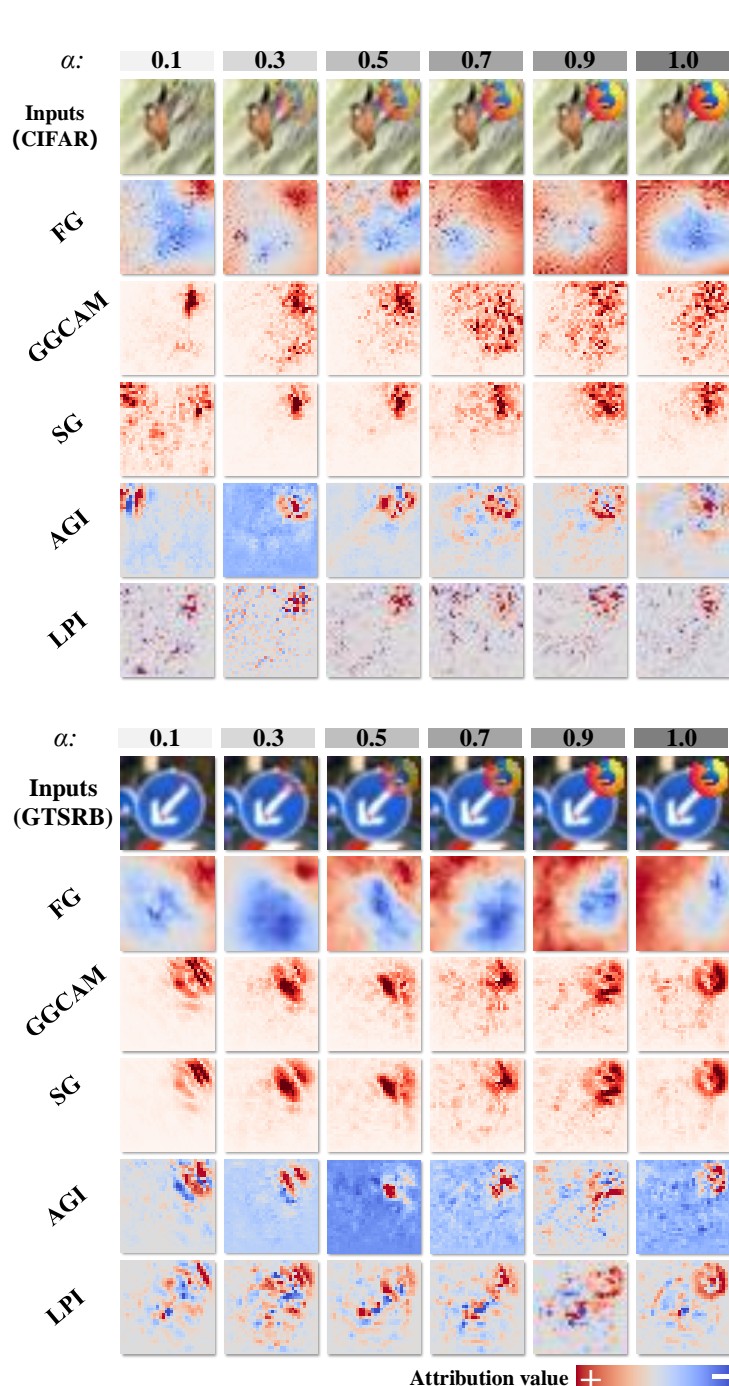

Figure 28: Visual comparison of attribution methods on CIFAR-10 and GTSRB datasets. Predictions made by ResNet-18 models Trojaned through the Blend attack with triggers of varying visibilities ($\alpha$) are explained using attribution methods including FG, GGCAM, SG, AGI and LPI.

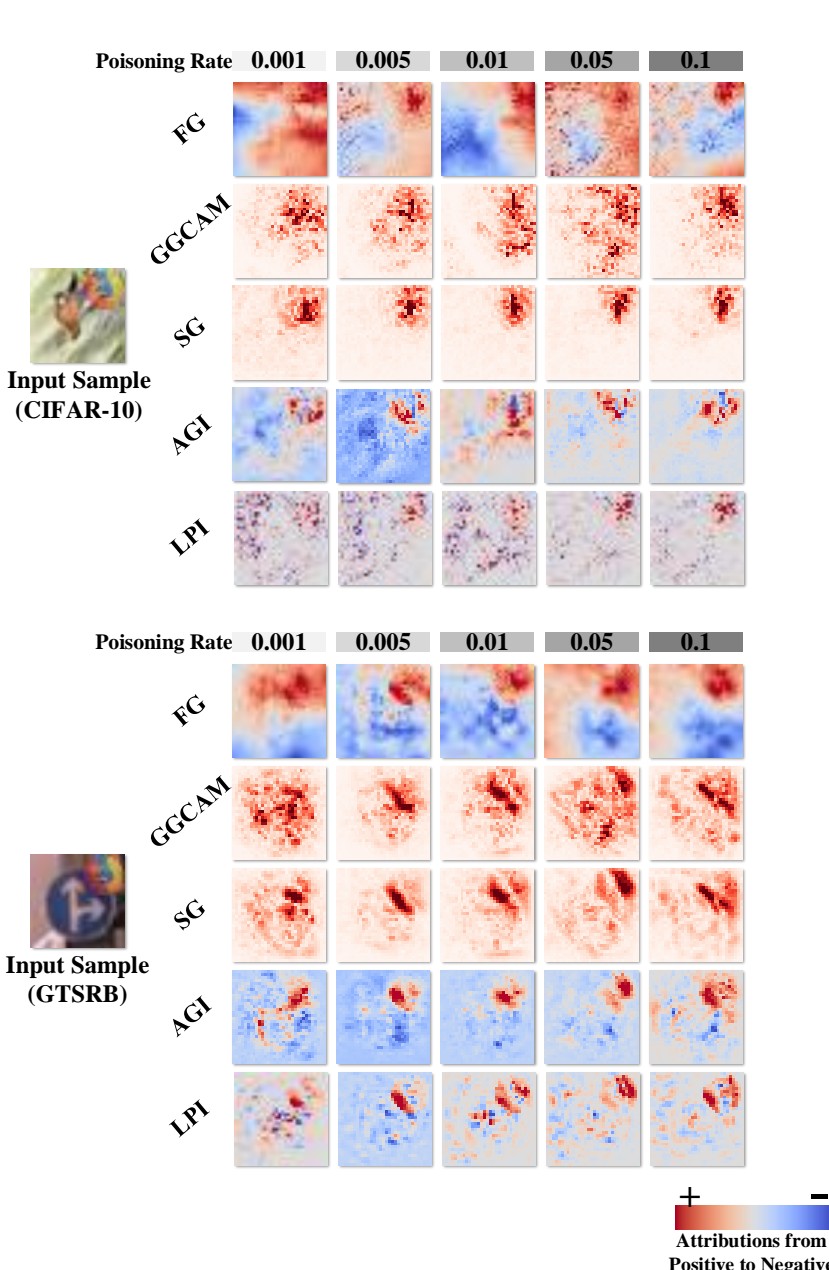

Figure 29: Visual comparison of attribution methods on CIFAR-10 and GTSRB datasets. Predictions made by ResNet-18 models Trojaned through the Blend attack of different poisoning rates are explained using three groups of attribution methods including FG, GGCAM, SG, AGI and LPI.

