# OpenReview forum: "A Backdoor-based Explainable AI Benchmark for High Fidelity Evaluation of Attributions"
_ICLR.cc/2026/Conference — ICLR 2026 Conference Withdrawn Submission_

### Official Review · Reviewer_cttG · 2025-10-29

**Soundness:** 3
**Presentation:** 3
**Contribution:** 3
**Rating:** 6
**Confidence:** 4

**Summary:**

This is an interesting and solid work. The authors propose a novel high-fidelity benchmark (BackX) based on backdoor attacks, along with a set of metrics for systematically evaluating attribution methods. The study is comprehensive and the design is innovative. However, the theoretical explanations and some experimental validations could be strengthened.

**Strengths:**

1. The research area is highly interesting and possesses significant practical application value.
2. The writing exhibits a logical flow of ideas, and the notation is clearly defined.
3. The research is comprehensive and robust. It proposes four essential criteria that a high-quality benchmark should fulfill and demonstrates that BackX largely meets these criteria. Subsequently, an extensive evaluation of various attribution methods is conducted based on this foundation.
4. The benchmark design is innovative. It cleverly utilizes backdoor attacks to generate controllable attribution ground truth, thereby effectively addressing the critical challenge of the lack of ground truth in attribution evaluation benchmarks.

**Weaknesses:**

1. The four fidelity criteria introduced in Section 2 are explained at a highly abstract level, making them difficult to grasp. For instance, the example `M = M_X + M_Y`used to illustrate "Attribution Verifiability" is confusing. The subsequent explanation seems to focus on preventing the mask S from introducing extra mutual information with the class, which makes the purpose of the `M_X + M_Y` decomposition unclear. It would be helpful to explain these concepts using concrete examples from existing XAI benchmarks.
2. Regarding the proof for Proposition 1 in Section 3.3, the condition `ε̃ ≤ ε` seems like a strong assumption. After adding a conspicuous trigger (such as Blend), the difference between the resulting sample `x̃` and the original sample `x` could be quite significant (judging from Figure 9, it is at least easily distinguishable to the human eye), so this condition may not necessarily hold. It is recommended to supplement the theory with experimental validation demonstrating that the recovered sample `x̂` is indeed close to `x`. The same concern applies to Proposition 2. If such experimental validation could be added, it would greatly enhance the persuasiveness of the paper, and I would recommend a higher score.
3. Concerning the proof for Input Distribution Invariance in Section 3.3, consider a scenario where the attribution method performs poorly or the Trojaned model itself is flawed. In such cases, the recovered sample `x̂`, generated using the mask `S^k`, might deviate more from the original sample `x` than the Trojaned sample `x̃` does. Would the core condition `Ω(x, x̂) ≤ ε̃`  still hold under these circumstances?

**Questions:**

1. Clarity of Fidelity Criteria: The four fidelity criteria in Section 2 are abstract. Please explain them more intuitively, for example, by using concrete cases from XAI benchmarks to illustrate `M = M_X + M_Y`.
2. Validation of Proposition 1: The condition `ε̃ ≤ ε` in Proposition 1 may be strong for conspicuous triggers (e.g., Blend). Could you add experiments to quantitatively show that the recovered sample `x̂`is indeed close to the original `x`?
3. Robustness of Input Distribution Invariance: The proof assumes the attribution mask `S^k` is effective. Please discuss if the condition `Ω(x, x̂) ≤ ε̃`  still holds if the attribution method performs poorly, and whether additional validation is needed.

---

### Official Review · Reviewer_RF8G · 2025-10-31

**Soundness:** 2
**Presentation:** 2
**Contribution:** 2
**Rating:** 2
**Confidence:** 5

**Summary:**

This paper proposes BackX, a benchmark for evaluating the faithfulness of XAI attribution methods. The core idea is to use models embedded with backdoor Trojans, where the implanted trigger serves as a controllable, verifiable "ground truth" for the model's prediction. The authors first define a set of four fidelity criteria (functional mapping invariance, input distribution invariance, attribution verifiability, and metric sensitivity) that they argue a reliable benchmark must satisfy. They then present a comprehensive evaluation setup, including a valuable analysis of confounding factors like post-processing choices (e.g., logits vs. probabilities). Using this benchmark, they evaluate a wide range of attribution methods on vision and text domains.

**Strengths:**

1.  Paper analyzes the impact of post-processing and output choice, is a valuable contribution.
2.  The evaluation on image-based tasks is thorough, covering multiple datasets, architectures, and a good variety of trigger types.

**Weaknesses:**

1.  My primary concern is that the core conceptual contribution of this work is limited. The idea of using model Trojans to create a ground truth for evaluating explanations is not new and builds directly on prior work (e.g., Lin et al., 2021). The paper's contribution feels more like an incremental consolidation and a more thorough implementation of this idea, rather than a fundamentally new paradigm. The "four foundational criteria," while a useful formalization, are largely a consolidation of concepts (like distribution shift) that are already widely discussed in the XAI evaluation literature.

2.  Flawed "Sole Ground Truth" Assumption & Contradictory NLP Evidence. The benchmark's entire validity rests on the assumption that the backdoor trigger is the *sole* causal factor for the model's prediction, and thus the *sole* ground truth attribution. This assumption is questionable and, more importantly, appears to be contradicted by the authors' own NLP experiments in Appendix A.11.

    * In Figure 20, the model is shown to attribute importance not only to the implanted trigger ("mn") but also to other semantically relevant sentiment words ("problem," "lacks").
    * This strongly suggests that the model's decision is *not* based solely on the trigger, but on a combination of the trigger and other input features.
    * If the true attribution is a combination of these features, then the benchmark's premise of using the trigger as the *sole* ground truth is flawed. This fundamentally undermines the benchmark's claim of "Attribution Verifiability" in this domain.

3.  The claim that this benchmark extends to the textual domain is not well-supported and feels like an afterthought. The evaluation is missing critical components:

    * The authors only test general-purpose, gradient-based methods (Grad, IG, SG, etc.) that are simply applied to token embeddings. There is a complete absence of attribution methods *specifically designed or optimized* for NLP (e.g., Integrated Directional Gradients (IDG), DiffMask, or attention-based methods). Without comparing against established, domain-specific techniques, the NLP evaluation lacks a meaningful baseline.
    * The authors did not provide a crucial control visualization for Figure 20: the attribution map for the *same sentence without the trigger*. Without this, it's impossible to know if the attribution on "problem" and "lacks" is standard behavior or an interaction effect with the trigger.

**Questions:**

See Weaknesses.

---

### Official Review · Reviewer_NGjV · 2025-11-01

**Soundness:** 2
**Presentation:** 3
**Contribution:** 2
**Rating:** 4
**Confidence:** 3

**Summary:**

The paper proposes a benchmark for measuring the faithfulness of models. The outline four important facets of measuring faithfulness (functional mapping invariance, input distribution invariance, attribution verifiability, and metric sensitivity), and highlight that theirs would be the first benchmark to cover (mostly) all points. Their benchmark works by taking a model with a backdoor, passing it samples with and without with the trigger in place, and measuring the explanation of both. The difference between explanations is used to identify the location of the trigger, which is then replaced with clean pixels from the original image. Faithfulness is measured as the ability to accurately perform this replacement using the explanation. The authors test the benchmark on a variety of different explanation techniques and find that there are notable differences in faithfulness across methods and settings.

**Strengths:**

The paper is clearly written and frames the considerations that should be taken when measuring faithfulness well. There is lots of discussion about the design choices of the benchmark, and the presentation overall is good. The benchmark is tested on different methods, and shows promise in measuring their faithfulness. I find the premise they propose for measuring the faithfulness clever and well thought out.

**Weaknesses:**

While I do see the value in this benchmark, I feel that the way it is currently framed in the paper paints it as more powerful than what I understand it to be. If these points can be addressed, I would be open to raising my score.

1. While the authors claim that their benchmark does not require a shift to the model, this is only true if the goal is to explain a model that has a backdoor. For models without this, the benchmark will require that a backdoor is added to them. This is a significant limitation of the method, and is not addressed in the paper. Personally, I don't feel that the claims of not changing the model are fully accurate because of this point.

2. The benchmark is *only* able to measure explanations of backdoor triggers. As synthetically inserted features, these triggers may differ from other features in how they are interpreted by both the model and the explanation method. Thus, I have concerns that measuring the faithfulness on these features may not be an accurate representation of the faithfulness overall.

3. It would be good to have more discussion of the backdoors used and their impact on the model and explanations. There is some vague discussion of how more visible triggers are not necessarily highlighted more strongly by the model that I feel relates in an important way to point #2. It would be helpful to see an expansion of this.

4. Regarding backdoors, I am also do not see sufficient evidence that the benchmark works equally well when using different backdoor methods. It would be useful to see some comparison of this.

5. There is not a lot of discussion regarding how well these results align (or don't) with prior benchmarking or analysis. It would be good to better situate the results.

**Questions:**

1. Do you have concerns regarding point #2 in weaknesses regarding how triggers may differ from "normal" features? Is this something you have observed or have evidence against?

2. Have you noticed an impact when using different backdoor methods?

3. How do these benchmark results compare to prior results?

---

### Official Review · Reviewer_uCAe · 2025-11-01

**Soundness:** 2
**Presentation:** 2
**Contribution:** 2
**Rating:** 4
**Confidence:** 4

**Summary:**

This paper introduces BackX, a new benchmark for evaluating the faithfulness of XAI attribution methods. The authors propose to solve the "lack of ground truth" problem by using backdoored models, where the implanted trigger serves as a verifiable, causal ground truth for a specific prediction. The paper first defines four "fidelity criteria" (functional mapping invariance, input distribution invariance, attribution verifiability, and metric sensitivity) that a good benchmark should possess. A significant part of the paper (Section 4) is dedicated to identifying and mitigating confounding factors, such as post-processing choices (e.g., `abs()` vs. original values) and the choice of output (logits vs. probabilities), to create a "standardized evaluation setup." The authors then apply this benchmark to a wide range of attribution methods on vision and (in the appendix) text datasets.

**Strengths:**

1.  The four fidelity criteria laid out in Section 2.1 are a valuable contribution. They provide a clear language and a useful lens through which to analyze and critique *all* XAI evaluation methods, not just this one.

2.  The authors have tested nine different attribution methods across three datasets (including the large-scale ImageNet) and (in the appendix) against five different attack types, including non-trivial ones like WaNet's geometric warping.

3.  Section 4 is perhaps the paper's strongest part. The analysis in Figures 3 and 4 convincingly demonstrates that many popular attribution methods are extremely sensitive to arbitrary post-processing choices.

**Weaknesses:**

1.  My main reservation is that the core idea feels more like a significant incremental improvement than a new paradigm. The use of backdoor triggers as a ground truth for XAI evaluation has been proposed before (e.g., Lin et al., 2021). While BackX is certainly a more comprehensive and well-formalized framework, it feels like an extensive validation and engineering effort built on a known concept, rather than a fundamental conceptual leap.

2.  The paper identifies that methods like Grad-CAM and FullGrad are incredibly sensitive to post-processing choices (e.g., explaining logits vs. probabilities). The paper's solution is to simply *pick one* configuration as the "standardized setup" and proceed with the benchmark. I would argue this is the wrong conclusion. The *real* finding here is not that we need a standard, but that these attribution methods themselves are not robust. If an explanation's fidelity score can be completely inverted by a simple `abs()` call or a `softmax()` operation, does this not imply the *method itself* is fundamentally unreliable, regardless of which setup you choose? The paper identifies this critical flaw but then sidesteps its implications, instead using it merely as a justification for standardization.

3.  The extension to text in Appendix A.11 feels tacked on and unconvincing. It only tests general-purpose gradient methods. It does not compare against any methods actually designed for NLP, which is a major omission for claiming cross-domain validity.

**Questions:**

1.  Given the extreme sensitivity to post-processing, shouldn't the main conclusion be that these methods are unreliable, rather than that we simply need a standardized setup?
2.  In Figure 20, the model also attributes importance to "problem" and "lacks." How do you reconcile this with the benchmark's assumption that the trigger is the *sole* ground truth?

---

### Note · Authors · 2025-11-12

I have read and agree with the venue's withdrawal policy on behalf of myself and my co-authors.